# Influence of Estuarine Tidal Mixing on Structure and Spatial Scales of Large River Plumes

Alexander Osadchiev[1,2,3], Igor Medvedev[1,4], Sergey Shchuka[1,3], Mikhail Kulikov[1], Eduard Spivak[5], Maria Pisareva[1], Igor Semiletov[5,6]

[1]Shirshov Institute of Oceanology, Russian Academy of Sciences, Moscow, Russia
[2]Institute of Geology of Ore Deposits, Petrography, Mineralogy and Geochemistry, Russian Academy of Sciences, Moscow, Russia
[3]Moscow Institute of Physics and Technology, Dolgoprudny, Russia
[4]Fedorov Institute of Applied Geophysics, Roshydromet, Moscow, Russia
[5]Ilyichov Pacific Oceanological Institute, Far Eastern Branch of the Russian Academy of Sciences, Vladivostok, Russia
[6]National Research Tomsk Polytechnic University, Tomsk, Russia

*Correspondence to*: Alexander Osadchiev (osadchiev@ocean.ru)

**Abstract.** The Yenisei and Khatanga rivers are among the largest estuarine rivers that inflow to the Arctic Ocean. Discharge of the Yenisei River is one order of magnitude larger than that of the Khatanga River. However, spatial scales of buoyant plumes formed by freshwater runoffs from the Yenisei and Khatanga gulfs are similar. This feature is caused by different tidal forcing in these estuaries, which have similar sizes, climate conditions, and geomorphology. The Khatanga discharge experiences strong tidal forcing that causes formation of a diluted bottom-advected plume in the Khatanga Gulf. This deep and weakly-stratified plume has a small freshwater fraction and, therefore, occupies a large area on the shelf. The Yenisei Gulf, on the other hand, is a salt-wedge estuary that receives a large freshwater discharge and is less affected by tidal mixing due to low tidal velocities. As a result, the low-salinity and strongly-stratified Yenisei plume has a large freshwater fraction and its horizontal size is relatively small. The results show that estuarine tidal mixing determines freshwater fraction in these river plumes, which governs their depth and area after they spread from estuaries to coastal sea. Therefore, influence of estuarine mixing on spatial scales of a large river plume can be of the same importance as the roles of river discharge rate and wind forcing. In particular, plumes with similar areas can be formed by rivers with significantly different discharge rates as illustrated by the Yenisei and Khatanga plumes.

## 1 Introduction

River plumes play an important role in land-ocean interactions. Despite their relatively small volume as compared to adjacent coastal seas, they significantly affect global fluxes of buoyancy, heat, terrigenous sediments, nutrients, and anthropogenic pollutants, which are discharged to the coastal ocean with continental runoff [Dagg et al., 2004; Milliman and Farnsworth, 2011; Lebreton et al., 2017; Schmidt et al., 2017]. As a result, dynamics and variability of river plumes are key factors for understanding mechanisms of spreading, transformation, and redistribution of continental discharge and river-

borne constituents in coastal seas and their influence on adjacent continental shelves [Geyer et al., 2004; Hickey et al., 2010; Hetland and Hsu, 2013]. World river plumes are characterized by wide variety of structure, morphology, and dynamical characteristics caused by large differences in regional features [Chant, 2011; Horner-Devine et al., 2015; Osadchiev and Korshenko, 2017; Osadchiev and Sedakov, 2019; Zavialov et al., 2020], in particular, estuarine conditions [Guo and Valle-Levison, 2007; Nash et al., 2009; Lai et al., 2016; Osadchiev, 2017].

River estuaries are areas where freshwater discharge initially interacts with saline sea water. The related processes of mixing of river runoff with sea water and formation of river plumes in estuaries determine their structure and govern their subsequent spreading and mixing in open sea. Intensity of estuarine mixing varies from negligible, when mostly undiluted freshwater discharge inflows directly to coastal sea, to dominant, which results in significant dilution of river discharge in well-mixed enclosed basins before being released to open sea [Schettini et al., 1998; Halverson and Palowicz, 2008; MacCready and Geyer, 2010; Geyer and MacCready, 2014].

The Yenisei and Khatanga rivers are among the largest estuarine rivers that flow into the Arctic Ocean (Fig. 1). These Yenisei and Khatanga gulfs are closely located and have similar sizes, geomorphology, and climatic conditions, albeit significantly different tidal forcing. In this study we focus on transformation of discharge of the Yenisei and Khatanga rivers in their estuaries and spreading of their buoyant plumes that occupy wide areas in the Kara and Laptev seas. Discharge of the Yenisei River is one order of magnitude larger than of the Khatanga River. However, spatial scales of buoyant plumes formed by freshwater runoffs from the Yenisei and Khatanga gulfs are similar. Using in situ hydrographic data, we reveal that this feature is caused by difference in intensity of estuarine tidal mixing that greatly affects spatial scales of these river plumes.

The paper is organized as follows. Section 2 provides the detailed information about the study area, the in situ, satellite, and wind reanalysis data used in this study. Section 3 describes the vertical structures and spatial extents of the Yenisei and Khatanga plumes, as well as tidal and wind forcing conditions in the study area. The influence of estuarine tidal forcing on spreading and mixing of the Yenisei and Khatanga plumes is analysed in Section 4 followed by the conclusions in Section 5.

## 2 Study area and data

### 2.1 Study area

Freshwater discharge from the Yenisei River (630 km$^3$ annually or 20000 m$^3$/s on average) is the largest among the Arctic rivers and accounts for 20% of total freshwater runoff to the Arctic Ocean [Gordeev et al., 1996; Carmack, 2000]. Hydrological regime of the Yenisei River is governed by a distinct freshet peak in June – July (half of total annual discharge), moderate discharge in May and August – September, and a drought in October – April [Pavlov et al., 1996; Guay et al., 2001]. The Yenisei River inflows into the Yenisei Gulf located at the southeastern part of the Kara Sea at the western side of the Taymyr Peninsula (Fig. 1a). The Yenisei Gulf is 250 km long; its width is 35-50 km. Average depth of the

southern (inner) part of the gulf increases off the river mouth from 5 to 15 m. The large Sibiryakov Island is located in northern (outer) part of the gulf and divides it into two 40-50 km long and 30-35 km wide straits (Fig. 1b). The western strait between the Sibiryakov and Oleniy islands is shallow (10 m deep), while depth of the eastern strait between the Sibiryakov Island and the Taymyr Peninsula steadily increases towards the open sea to 25-30 m and connects the Yenisei Gulf with the central part of the Kara Sea. The Yenisei Gulf is covered by ice in October – July.

Freshwater discharge from the Khatanga River (105 km$^3$ annually or 3300 m$^3$/s on average) is much smaller than that from the Yenisei River. Approximately one half of this volume is discharged to the Laptev Sea during a freshet period in June and then the river discharge steadily decreases till September [Pavlov et al., 1996]. The lower part of the Khatanga River is completely frozen in October – May and river discharge is negligible during this period [Pavlov et al., 1996]. The Khatanga River inflows into the Khatanga Gulf which is located at the southwestern part of the Laptev Sea at the eastern side of the Taymyr Peninsula (Fig. 1a). Shape, bathymetry, spatial scales, and climatic conditions of the Khatanga Gulf are similar to those of the Yenisei Gulf which is located approximately 800 km to the west from the Khatanga Gulf. The Khatanga Gulf is 250 km long; its width is 25-50 km. The shallow (5-20 m deep) inner and deep (20-30 m deep) outer parts of the gulf are connected by a narrow strait (15-20 km wide) between the Khara-Tumus and Taymyr peninsulas (Fig. 1c). The Bolshoy Begichev Island is located in the outer part of the Khatanga Gulf and divides it into two straits. The southern strait is narrow (8 km wide) and shallow (10 m deep), while depth of the northern strait (15-20 km wide) between the Bolshoy Begichev Island and the Taymyr Peninsula steadily increases towards the open sea to 25-30 m and connects the Khatanga Gulf with the western part of the Laptev Sea. The Khatanga Gulf is covered by ice from October to July-August. Tides in the Khatanga Gulf are among the largest in the Eurasian part of the Arctic Ocean [Pavlov et al., 1996; Kulikov et al., 2018].

## 2.2 Data

Hydrographic in situ data used in this study were collected during three oceanographic field surveys in the Kara and Laptev seas including the 4th cruise of the R/V "Nikolay Kolomeytsev" on 27-29 August 2000 in the southwestern part of the Laptev Sea, the 66th cruise of the R/V "Akademik Mstislav Keldysh" on 24-26 July 2016 in the Yenisei Gulf and the central part of the Kara Sea, the 69th cruise of the R/V "Akademik Mstislav Keldysh" on 17-18 September 2017 in the Khatanga Gulf and the southwestern part of the Laptev Sea (Fig. 1). Field surveys included continuous measurements of salinity in the surface sea layer (2-3 m depth) performed along the ship track using a ship-board pump-through system equipped by a thermosalinograph (*Sea-Bird Electronics SBE 21 SeaCAT*). Vertical profiles of salinity were performed using a conductivity-temperature-depth (CTD) instrument (*Sea-Bird Electronics SBE 911plus*) at 0.2 m vertical resolution. This CTD profiler was equipped with two parallel temperature and conductivity sensors; the mean temperature differences between them did not exceed 0.01°C, while differences of salinity were not greater than 0.01.

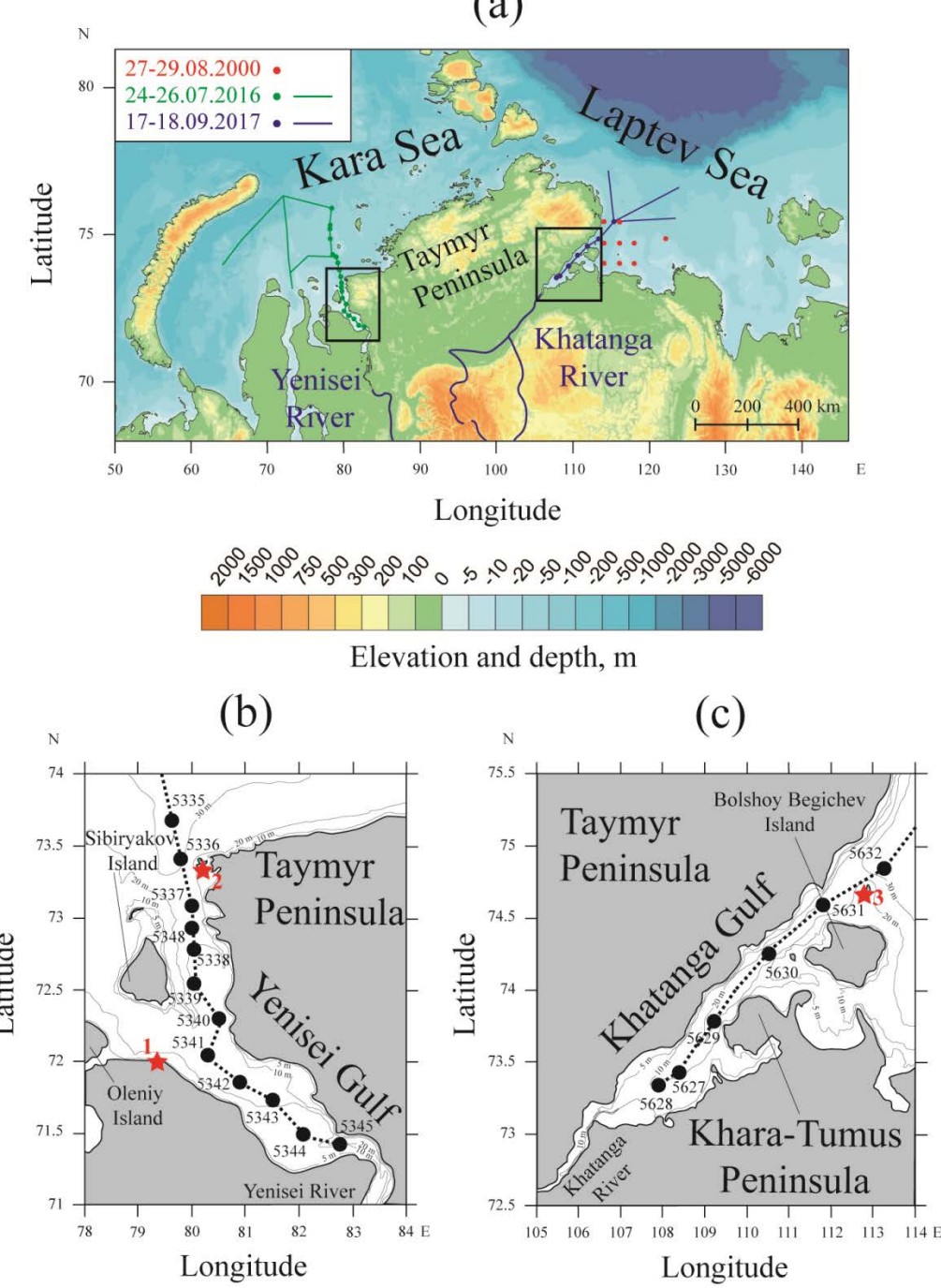

Figure 1: The bathymetry of the Kara and Laptev seas according to the IBCAO database; ship tracks (lines) and location of hydrographic stations (circles) of three oceanographic field surveys conducted in August 2000 (red) and September 2017 (blue) in the Laptev Sea and in July 2016 (green) in the Kara Sea (a). Ship tracks, and location of hydrographic stations (black circles) and tidal gauge stations (red stars: 1 – Leskin, 2 – Dikson, 3 – Preobrazheniye) in the Yenisei (b) and Khatanga (c) gulfs. The graphic scale in panel (a) corresponds to the latitude of 73°.

The tidal data used in this study includes in situ data from tidal gauge stations and results of tidal numerical modelling. We used hourly sea level records from two tide gauges located in the Yenisei Gulf, namely, Leskin and Dikson (Fig. 1b), and the Preobrazheniye tide gauge station in the Khatanga Gulf (Fig. 1c). Tidal gauge data were downloaded from the Unified State System of Information about the World Ocean (ESIMO) website (http://portal.esimo.ru/portal). Tidal velocity data in the study area were obtained from the Arctic Ocean Tidal Inverse Model (AOTIM5) [Padman and Erofeeva, 2004]. The AOTIM5 is based on the data assimilation scheme [Egbert et al., 1994] and presents an inverse solution assimilating almost all available tide gauge data in the Arctic Ocean [Padman and Erofeeva, 2004]. The assimilated data consist of coastal and benthic tide gauges (between 250 and 310 gauges per constituent), 364 cycles of TOPEX/Poseidon and 108 cycles of ERS altimeter data. The model bathymetry is based on the International Bathymetric Chart of the Arctic Ocean (IBCAO) [Jakobsson et al., 2000] reprojected on a uniform 5-km grid. Tidal simulations were run in absence of sea ice.

Wind forcing conditions were examined using ERA5 atmospheric reanalysis with a 0.25° spatial and hourly temporal resolution. The reanalysis data were downloaded from the European Centre for Medium-Range Weather Forecasts (ECMWF) website (https://www.ecmwf.int/en/forecasts/datasets/archive-datasets/reanalysis-datasets/era5). Discharge data used in this study was obtained at the most downstream gauge stations at the Yenisei and Khatanga rivers located approximately 650 and 200 km upstream from the river mouths, respectively. The river discharge data were downloaded from the Federal Service for Hydrometeorology and Environmental Monitoring of Russia (FSHEMR) website (http://gis.vodinfo.ru/).

Satellite data used in this study include Terra/Aqua Moderate Resolution Imaging Spectroradiometer (MODIS) satellite imagery provided by the National Aeronautics and Space Administration (NASA). MODIS L1b calibrated radiances including MODIS bands 1 (red), 3 (blue), 4 (green), and daytime 31 (thermal) were downloaded from the NASA web repository (https://ladsweb.modaps.eosdis.nasa.gov/). We used ESA BEAM software [Fomferra and Brockmann, 2005] for retrieving maps of sea surface distributions of corrected reflectance (CR), concentration of chlorophyll-a (Chl-a), and brightness temperature (BT) at the study areas with spatial resolutions of 500 m, 500 m, and 1 km, respectively. Chl-a distributions were calculated using the Ocean Color 3M (OC3M) algorithm [O'Reilly et al., 1998; Werdell and Bailey, 2005]. Due to the complexity of coastal processes that govern the temperature of the sea surface and the absence of specific regional algorithms for retrieving sea surface temperature (SST) from satellite data in the study areas with very limited in situ measurements, we did not use the standard SST product of MODIS. Instead, we used a BT product that does not provide an accurate temperature of the sea surface, but shows relative temperature differences, which can be used to detect spreading areas of the river plumes. Several previous studies showed that elevated surface temperature and increased concentrations of Chl-a are stable characteristics of the large Arctic river plumes which form distinct frontal zones with cold and low-fluorescent ambient sea [Glukhovets and Goldin, 2014; Kubryakov et al., 2016; Glukhovets and Goldin, 2019; Osadchiev et al., 2020].

**3 Results**

**3.1 Tidal forcing in the Yenisei and Khatanga gulfs**

To study tidal forcing in the Yenisei and Khatanga gulfs we analysed in situ data of sea level measured at three tide gauges, namely: Leskin and Dikson in the Yenisei Gulf (Fig. 1b), Preobrazheniye in the Khatanga Gulf (Fig. 1c). These measurements were performed every hour during summer and autumn at the Leskin station (1977 – 1979, 1981 – 1992) and during the whole year at the Dikson (1977 – 1979, 1981 – 1990) and Preobrazheniye (1978 – 1979, 1981 – 1988) stations. The tidal harmonic constants were calculated from these data by the *Matlab* harmonic analysis toolbox *T_Tide* [Pawlowicz et al., 2002]. The periods of tidal measurements were split to one year long time series, for these series we estimated the vector averaged amplitudes. Then the obtained amplitudes from different years were averaged to calculate the mean long-term values of the amplitudes of the tidal harmonics [Medvedev et al. 2013]. As a result, we estimated amplitudes of 68 tidal constituents including overtides (multiple tidal harmonics) for the Dikson and Preobrazheniye stations, as well as amplitudes of 35 or 50 tidal constituents (for different years) for the Leskin station. Tidal circulation in the study areas is dominated by the main semidiurnal tidal constituent ($M_2$) and is also affected by the $S_2$, $K_1$, and $O_1$ constituents, while the role of the other tidal harmonics is insignificant. The obtained mean amplitudes of four major tidal constituents and the mean spring tidal ranges at the considered stations are presented in Table 1.

Table 1: The mean amplitudes of four major tidal constituents and the mean spring tidal ranges reconstructed for the Leskin, Dikson, and Preobrazheniye tide gauge stations.

| Tidal gauge stations | Mean amplitudes, cm | | | | Mean spring tidal ranges, cm |
|---|---|---|---|---|---|
| | $M_2$ | $S_2$ | $K_1$ | $O_1$ | |
| Leskin | 16.0 | 7.3 | 3.4 | 1.7 | 46.5 |
| Dikson | 8.2 | 3.9 | 1.9 | 1.2 | 24.2 |
| Preobrazheniye | 35.1 | 15.1 | 5.0 | 2.7 | 100.2 |

The smallest mean amplitudes of the main tidal harmonic $M_2$ and mean spring tidal ranges ($2M_2 + 2S_2$) are observed at the Dikson station (8.2 and 24.2 cm) at the mouth of the Yenisei Gulf. The mean tidal amplitudes and ranges are twice larger (16.0 and 46.5 cm) in the Leskin station between the inner and outer parts of the Yenisei Gulf, and four times larger (35.1 and 100.2 cm) at the Preobrazheniye station at the mouth of the Khatanga Gulf, as compared to the Dikson station. These results are in good agreement with previous assessments of tidal forcing in the Yenisei and Khatanga gulfs [Voinov, 2002; Korovkin and Antonov, 1938] and demonstrate that tidal forcing in the Khatanga Gulf is significantly stronger than in the Yenisei Gulf. Moreover, Korovkin and Antonov [1938] reported that tidal forcing intensifies in the outer part of the Khatanga Gulf. In situ measurements performed in August 1934 showed that the amplitude of $M_2$ increased from 42 cm near the Preobrazheniye Island (where the Preobrazheniye station is located) to 50 cm at the western shore of the Bolshoy Begichev Island and to 83 cm near the Khara-Tumus Peninsula (that separates the inner and outer parts of the gulf)

[Korovkin and Antonov, 1938]. The registered spring tidal range near the western shore of the Bolshoy Begichev Island was 150 cm and increased to 259 cm near the Khara-Tumus Peninsula.

The amplitude of $M_2$ and spring tidal range at the Preobrazheniye station have large seasonal and inter-annual variability caused by variability of river discharge and ice coverage. In particular, the reconstructed mean annual amplitudes of $M_2$ varied from 32 to 42 cm, while the mean monthly spring tidal range varied from 87 cm in May to 130 cm in August [Kulikov et al., 2020]. Therefore, in order to analyse influence of tidal mixing on structure and spreading of the Yenisei and Khatanga plumes during the periods of field measurements addressed in this study, we reconstructed velocities of tidal currents in the Yenisei Gulf in July 2016 and in the Khatanga Gulf in September 2017 using the AOTIM5 tidal model (Fig. 2). Voinov [2002] reported that maximal velocities of tidal currents in the Yenisei Gulf in summer are equal to 30–40 cm/s in the outer part of the gulf along the Taymyr Peninsula. The modelled maximal tidal velocities in July 2016 were equal to 10-20 cm/s in the inner part of the estuary and up to 25 cm/s in the outer part of the estuary (Fig. 2a). Modelled maximal tidal velocities in the Khatanga Gulf were much larger than in the Yenisei Gulf and increased from 20–50 cm/s in the outer part of the estuary to 80–100 cm/s in the inner part of the estuary (Fig. 2a).

Basing on tidal data simulated by AOTIM5 model and IBCAO bathymetric data we calculated distribution of the Simpson-Hunter parameter $K = h\ /\ U^3$, where $h$ is the sea depth and $U$ is the average tidal velocity [Simpson and Hunter, 1974]. This parameter is indicative of intensity of tidal-induced mixing of water column [Simpson and Hunter, 1974; Garrett et al., 1978] and was used in many studies for identification of sea areas where tidal-induced turbidity penetrates from sea bottom to the surface layer [Chen et al., 2009; Korotenko et al., 2014]. In this study we used tidal velocities averaged in the Yenisei Gulf during July 2016 and in the Khatanga Gulf during September 2017 in the following way: $\overline{U} = \langle \sqrt{(u^2 + v^2)} \rangle$ , where $u$ and $v$ are the time series of the north and east components of the tidal currents reconstructed from the eight tidal constituents ($M_2$, $S_2$, $K_1$, $O_1$, $N_2$, $K_2$, $P_1$, $Q_1$). Due to wide ranges of variability of $K$ in the Yenisei and Khatanga gulfs we analyzed its common logarithm $\log_{10}(K)$ which distribution is shown in Fig. 2b. The value of $\log_{10}(K)$ is relatively large (3.5–5) in the Yenisei Gulf that shows low influence of tides on estuarine mixing in the surface layer. In contrast to the Yenisei Gulf, the value of $\log_{10}(K)$ is near 1.5–3 within the majority of area of the Khatanga Gulf that indicates intense tidal-induced mixing in this area (Fig. 2b).

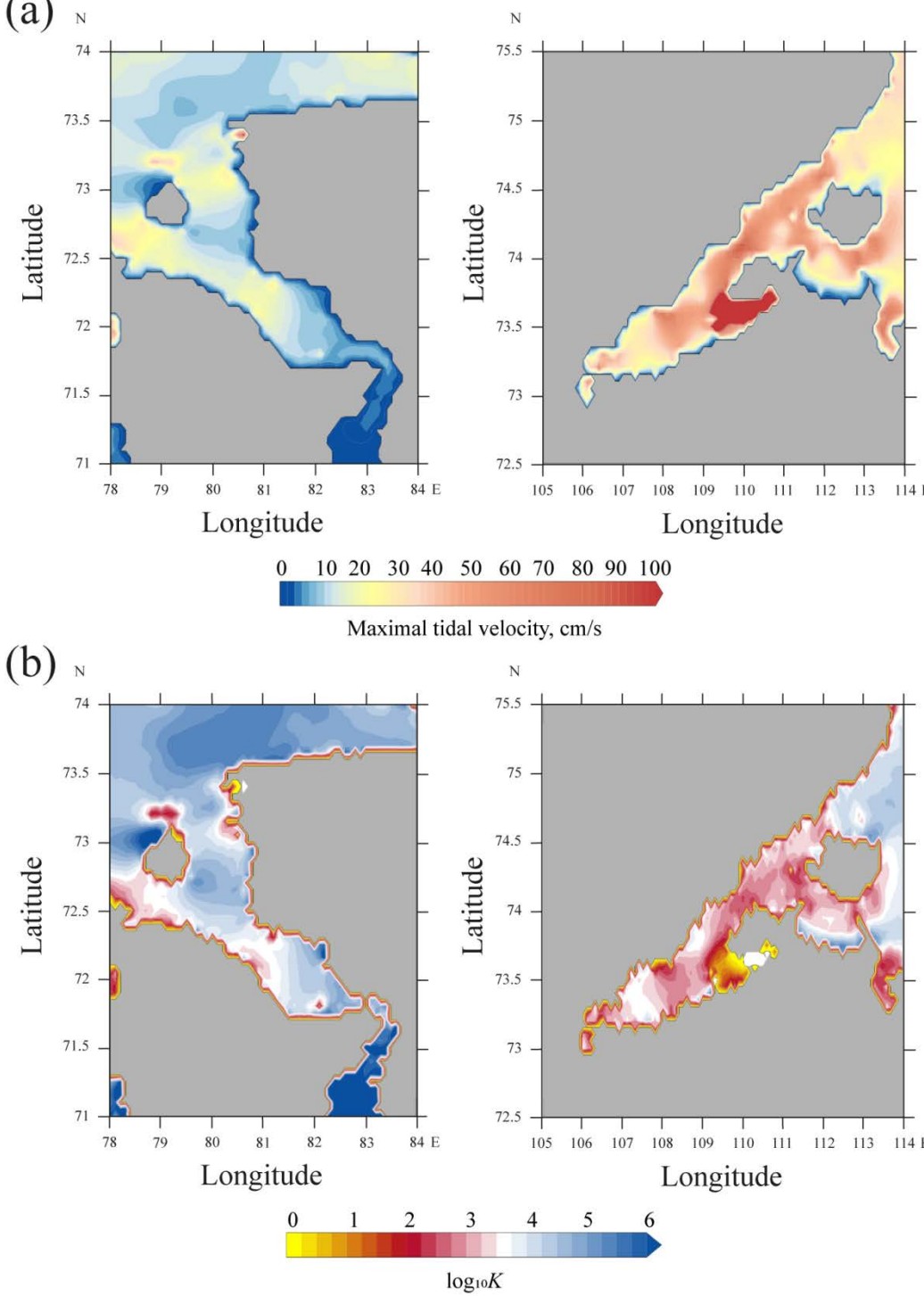

**Figure 2: Distributions of maximal tidal velocity (a) and common logarithm of the Simpson-Hunter parameter *K* (b) in the Yenisei Gulf in July 2016 (left) and in the Khatanga Gulf in September 2017 (right).**

## 3.2 Wind forcing in the Kara and Laptev seas

Using the ERA5 atmospheric reanalysis, we studied influence of wind forcing on the Yenisei and Khatanga plumes during the periods preceding in situ measurements in 2016 and 2017. For this purpose, we reconstructed daily averaged wind speed and direction during 29 June – 26 July 2016 in the central part of the Kara Sea (Fig. 3) and during 8 August – 18 September 2017 in the southwestern part of the Laptev Sea (Fig. 4). These wind time series cover ice-free periods at the study areas from decline of ice coverage to in situ measurements in the Yenisei and Khatanga plumes, i.e., the periods when wind forcing affected spreading of the river plumes.

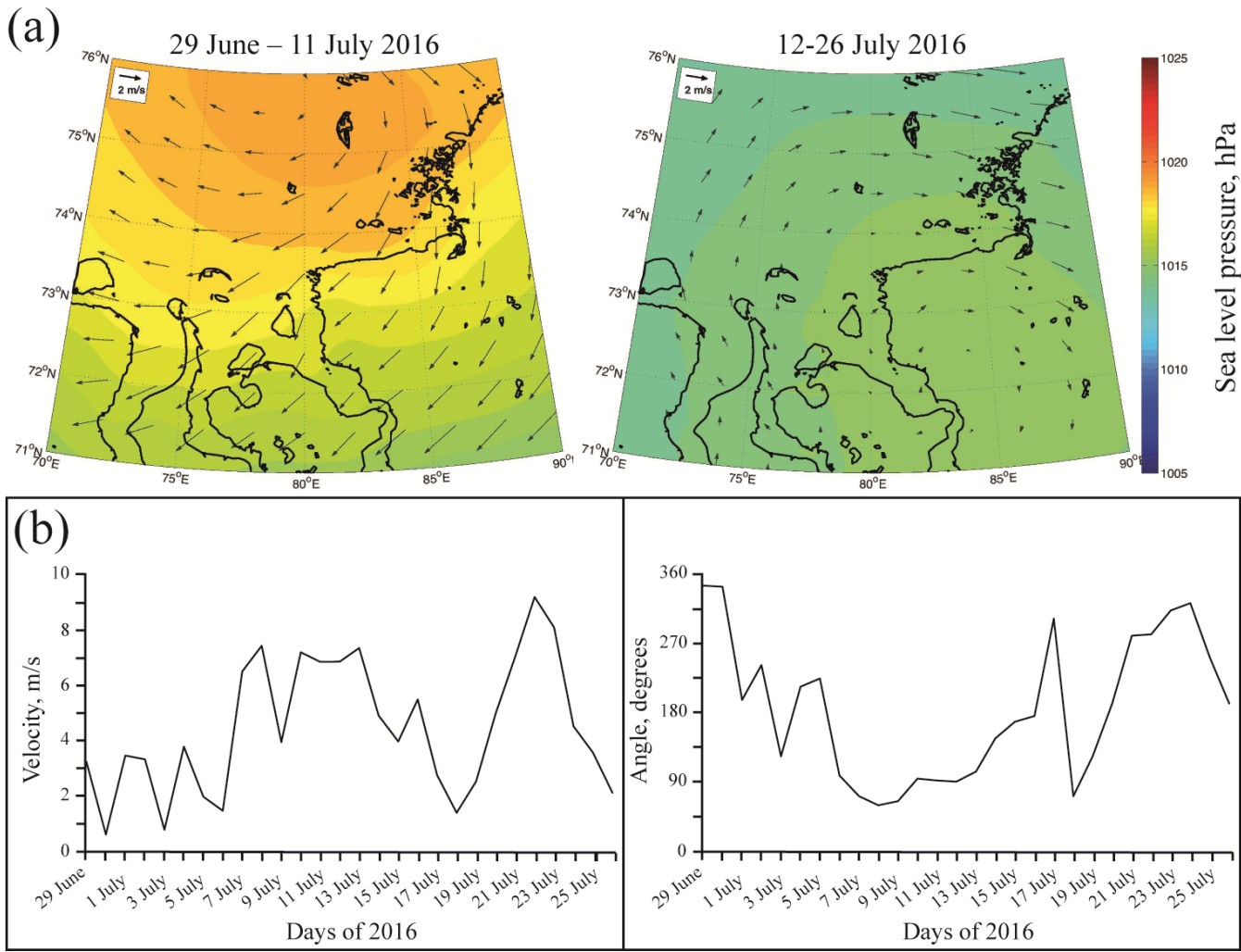

**Figure 3: Average wind forcing (arrows) and sea level pressure (color) during 29 June – 11 July (left) and 12-26 July 2016 (right) in the central part of the Kara Sea (a). Daily average wind speed (left) and direction (right) during 29 June – 26 July 2016 (b). Wind data were obtained from the ERA5 atmospheric reanalysis.**

Wind forcing in the central part of the Kara Sea was weak and moderate for the majority of days during 29 June – 26 July 2016 (Fig. 3). Daily averaged values of wind speed at the study area varied between 1 and 9 m/s, while their mean value was 4 m/s (Fig. 3b). On 10-13 July 2016, i.e., during 4 days, the longest continuous period of strong winds (> 5 m/s) was observed in the central part of the Kara Sea. Moreover, the direction of predominant winds showed significant synoptic variability during the considered period due to high variability of atmospheric pressure. As a result, wind forcing averaged during 2-week time periods is characterized by even smaller wind speed, namely, < 4 m/s during 29 June – 11 July and < 2 m/s during 12-26 July, (Fig. 3a).

Similarly to the central part of the Kara Sea in July 2016, wind forcing in the southwestern part of the Laptev Sea was mainly low/moderate and highly variable during 8 August – 18 September 2017 (Fig. 4). No storm events occurred during this period, daily averaged values of wind speed varied between 1 and 8 m/s, while their mean value was 4 m/s (Fig. 4b). The longest continuous period of strong wind forcing lasted only during 3 days (6-8 September 2017). Atmospheric circulation in the southwestern part of the Laptev Sea was dominated by multiple cyclones and anticyclones during the considered period. Due to the observed intraday variability of predominant wind direction, speed of 2-week averaged wind forcing was < 2 m/s during 8-21 August 2017 and was < 4 m/s during 22 August – 3 September and 4-18 September 2017 (Fig. 4a).

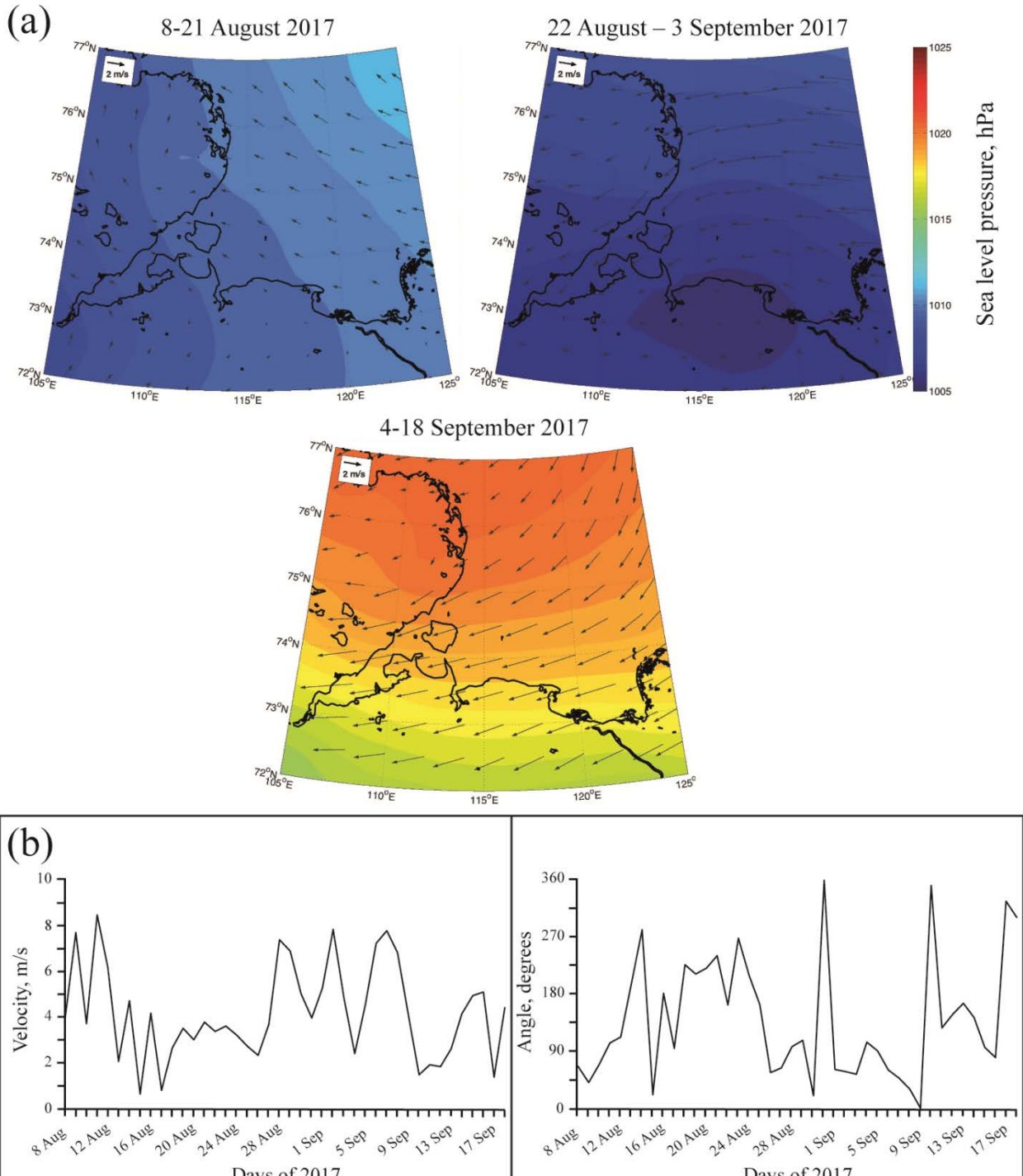

**Figure 4: Average wind forcing (arrows) and sea level pressure (color) during 8-21 August (top left), 22 August – 3 September (top right), and 4-18 September 2017 in the western part of the Laptev Sea (a). Daily average wind speed (left) and direction (right) during 8 August – 18 September 2017 (b). Wind data were obtained from the ERA5 atmospheric reanalysis.**

### 3.3 Vertical salinity structure of the Yenisei and Khatanga plumes

The in situ measurements performed in July 2016 in the Kara Sea during the period of peak discharge of the Yenisei River (approximately 30000 m$^3$/s) revealed that the Yenisei plume was spreading more than 500 km northward from the river mouth and its depth and vertical salinity structure did not change much along this distance (Fig. 5a). In order to quantify the effect of estuarine mixing on the Yenisei plume, we calculated vertical distribution of freshwater fraction $F = (S_0 - S) / S$ along the transect (Fig. 5b), where $S$ is the observed salinity, $S_0$ is the reference ambient sea salinity prescribed equal to 32. $F$ is equal to the ratio between the volumes of river water and sea water that were mixed and produced the observed salinity in the water parcel of the Yenisei plume. The value of the reference salinity equal to 32 was chosen according to typical salinity of ambient sea water at the shelves of the central part of the Kara Sea and the southwestern part of the Laptev Sea [Pavlov et al., 1996; Johnson et al., 1997; Williams and Carmack, 2015]. In order to assess the process of dilution of freshwater discharge within the Yenisei plume, we defined five different salinity ranges of the river plume water, namely, $0 < S < 5$, $5 < S < 10$, $10 < S < 15$, $15 < S < 20$, $20 < S < 25$, as well as the salinity range $S > 25$ for the ambient sea. Then for all vertical salinity profiles of the transect we calculated the local shares of freshwater "volume" in water column among these salinity ranges, i.e., what percentage of total freshwater volume contained in the water column is located between the isohalines of 0 and 5 (salinity range of 0-5), between the isohalines of 5 and 10 (salinity range of 5-10), etc. (Fig. 5c). Note, that these freshwater fractions are not strictly volumes because they are based on the measurements along the estuary and do not consider the varying width of the estuary.

Figure 5 illustrates that the Yenisei discharge experienced relatively little mixing in the estuary due to weak tidal forcing. The Yenisei plume occupied the whole water column in the shallow inner part of the estuary, its surface salinity was 0-5 (stations 5341-5345). Further northward the plume detached from the sea bottom. The plume depth (defined by the isohaline of 25) remained equal to 8-12 m in the outer part of the estuary (stations 5337-5340 and 5348) and at the Kara Sea shelf (stations 5333-5336 and 5349-5350). Surface salinity of the plume slowly increased from 6 to 10 in this 300 km long part of the transect. Sharp salinity gradient was observed between the plume and the subjacent sea, vertical distance between the isohalines of 10 and 25 was 2-5 meters. Freshwater fraction remained concentrated in the shallow and low-salinity surface layer in the open part of the Kara Sea till station 5350 located 500 km from the river mouth. The majority of freshwater "volume" in water column was located in two salinity ranges, namely, in 0-5 salinity range in the inner estuary and in 5-10 salinity range in the outer estuary and at the sea shelf (Fig. 5c). Further northward surface salinity abruptly increased from 15 (station 5351) to 28 (station 5352) in a distance of 10 km indicating the northern boundary of the Yenisei plume.

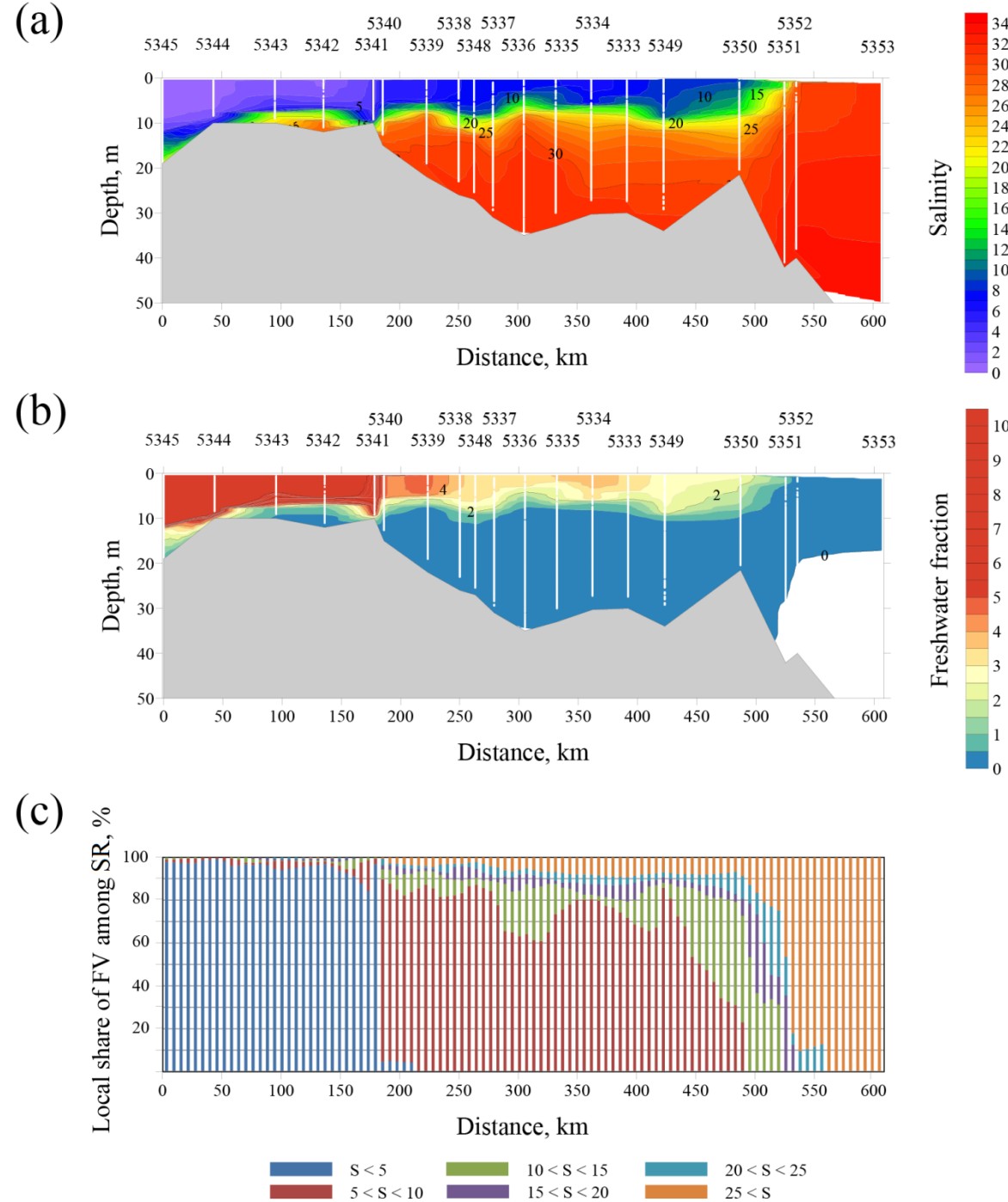

Figure 5: The vertical salinity structure (a), distribution of freshwater fraction (b), local shares of freshwater "volume" (FV) among salinity ranges (SR) in water column (c) along the ship track in the Yenisei Gulf and the adjacent shelf of the Kara Sea on 24-26 July 2016.

Despite low discharge rate of the Khatanga River (approximately 3000 m$^3$/s) in September 2017, the horizontal extent of the Khatanga plume during 17-18 September 2017 was similar to that of the Yenisei plume, while its maximal depth even

exceeded the depth of the Yenisei plume (Fig. 6a). The Khatanga discharge experienced intense estuarine tidal mixing and, therefore, was distributed from surface to bottom in the inner estuary (stations 5627-2629) and over 20-25 m deep water column in the outer estuary (Fig. 6b). Tidal-induced dilution caused increase of surface salinity and depth of the plume from 4 and 7 m (station 5628) to 17 and 25 m (station 5630) at 120 km along the transect. In the outer part of the estuary the plume detached from sea bottom and its depth steadily decreased to 11 m, while surface salinity increased to 21 (station

5632). Further northeastward at the Laptev Sea shelf the plume salinity slightly increased to 22, while depth slightly decreased to 9 m in a distance of 100 km from the Khatanga Gulf (station 5591). Freshwater fraction in the Khatanga plume also slightly changed while it was spreading from the estuary to the open part of the Laptev Sea. Freshwater "volume" in the water column steadily transferred from 0-5 salinity range near the river mouth to 20-25 salinity range in the open sea (Fig. 6c). Vertical distance between the isohalines of 25 and 30 was 2-4 meters in the outer estuary and the shelf sea.

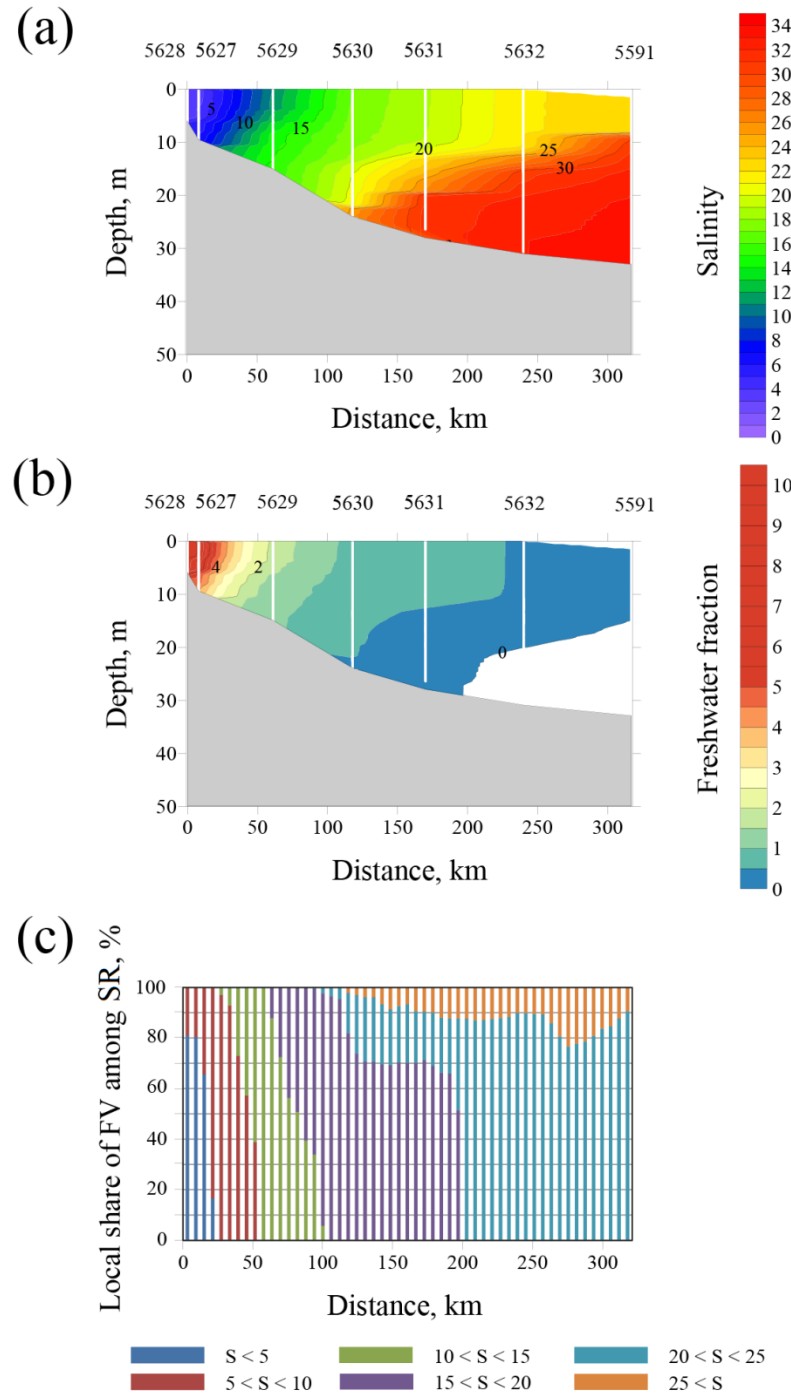

**Figure 6: The vertical salinity structure (a), distribution of freshwater fraction (b), local shares of freshwater "volume" (FV) among salinity ranges (SR) in water column (c) along the ship track in the Khatanga Gulf and the adjacent shelf of the Laptev Sea on 17-18 September 2017.**

## 3.4 Spatial extents of the Yenisei and Khatanga plumes

In order to assess and compare spatial extents of the Yenisei and Khatanga plumes, we analysed in situ salinity measurements in the surface layer performed in the Kara and Laptev seas during field surveys referred above (Fig. 7), as well as satellite observations of the study areas in 2000-2019 (Fig. 8-10). Continuous surface salinity measurements along the ship track detected the border of the river plume in the central part of the Kara Sea in July 2016 defined by the isohaline of 25 (Fig. 7a). This buoyant plume was formed by discharges of the Ob and Yenisei rivers, i.e., river runoff from the closely located Ob and Yenisei gulfs formed the Ob and Yenisei plumes that coalesced into the joint Ob-Yenisei plume [Pavlov et al., 1996; Zatsepin et al., 2010; Zavialov et al., 2015].

The meridional transects from the Ob and Yenisei gulfs in July 2016 (Fig. 7a) gave two crossings of the northern boundary of the Ob-Yenisei plume. Surface salinity measurements along the other segments of the ship track showed absence of the Ob-Yenisei plume in the western and northern parts of the Kara Sea. However, the detailed location of the boundary of the Ob-Yenisei plume was not detected. Also, the performed salinity measurements did not distinguish the western part of the Ob-Yenisei plume formed by the Ob discharge and the eastern part of the Ob-Yenisei plume formed by the Yenisei discharge. These drawbacks of in situ measurements were substituted by satellite observations of the study area.

Due to common cloudy weather conditions in the Kara Sea, we analysed the satellite imagery of the study area acquired on 15 July 2016 when the Ob-Yenisei plume was clearly seen in optical satellite images and the structure of surface turbidity, temperature, and chlorophyll-a could be identified (Fig. 8). Satellite Chl-a distribution in the Kara Sea reveals location of the border of the Ob-Yenisei plume (Fig. 8c) which is in a very good accordance with in situ salinity measurements performed 9-11 days after the satellite observations (Fig. 7a). Significant differences in temperature and concentration of suspended sediments in the Ob (10-12 °C and 40 $g/m^3$) and Yenisei (16-18 °C and 10 $g/m^3$) river water [Gordeev et al., 1996] provides opportunity to distinguish the Ob- and Yenisei-dominated parts of the Ob-Yenisei plume.

CR (Fig. 8a) and BT (Fig. 8b) satellite products show, first, the meandering turbid and cold jet from the Gulf of Ob that forms the relatively small western Ob-dominated part of the Ob-Yenisei plume and, second, the low-turbid and warm jet from the Yenisei Gulf that forms the large warm and low-turbid eastern Yenisei-dominated part of the Ob-Yenisei plume. The observed difference in areas of the Ob- and Yenisei-dominated parts is caused by different discharge regimes of the Ob and Yenisei rivers. The discharge rate of the Yenisei River has distinct freshet peak in June – July, while the discharge rate of the Ob River is characterised by steady increase and decrease of discharge in May – September [Pavlov et al., 1996; Osadchiev et al., 2017]. As a result, the average freshwater runoff from the Yenisei River in June – July (~ 60 000 $m^3/s$) is twice greater than that from the Ob River (~ 30 000 $m^3/s$). Therefore, based on the joint analysis of in situ and satellite data, we detected the spreading area of the Yenisei plume in July 2016 and revealed that its boundary was located 200-250 km from the Yenisei Gulf.

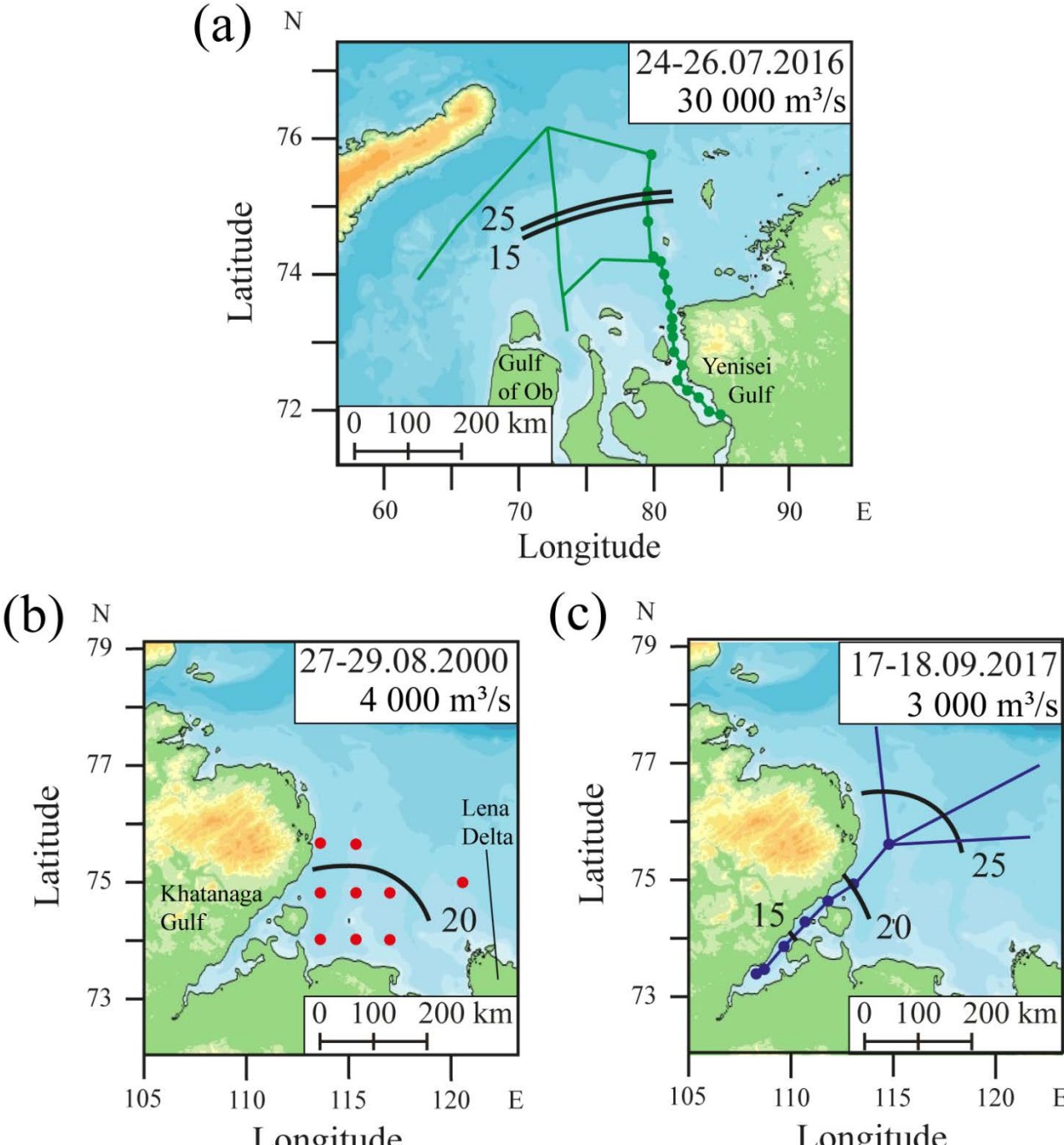

**Figure 7: Location of the isohalines of 15, 20, and 25 at the surface layer (black lines) at the central part of the Kara Sea (a) and the western part of the Laptev Sea (b, c) indicating spatial scales of the Ob-Yenisei (a) and Khatanga (b, c) river plumes reconstructed from in situ measurements conducted in August 2000 (red) and September 2017 (blue) in the Laptev Sea and in July 2016 (green) in the Kara Sea. The dates of the field measurements and the discharge rates of the rivers during these periods are shown at the figures. The graphic scales correspond to the latitude of 73°.**

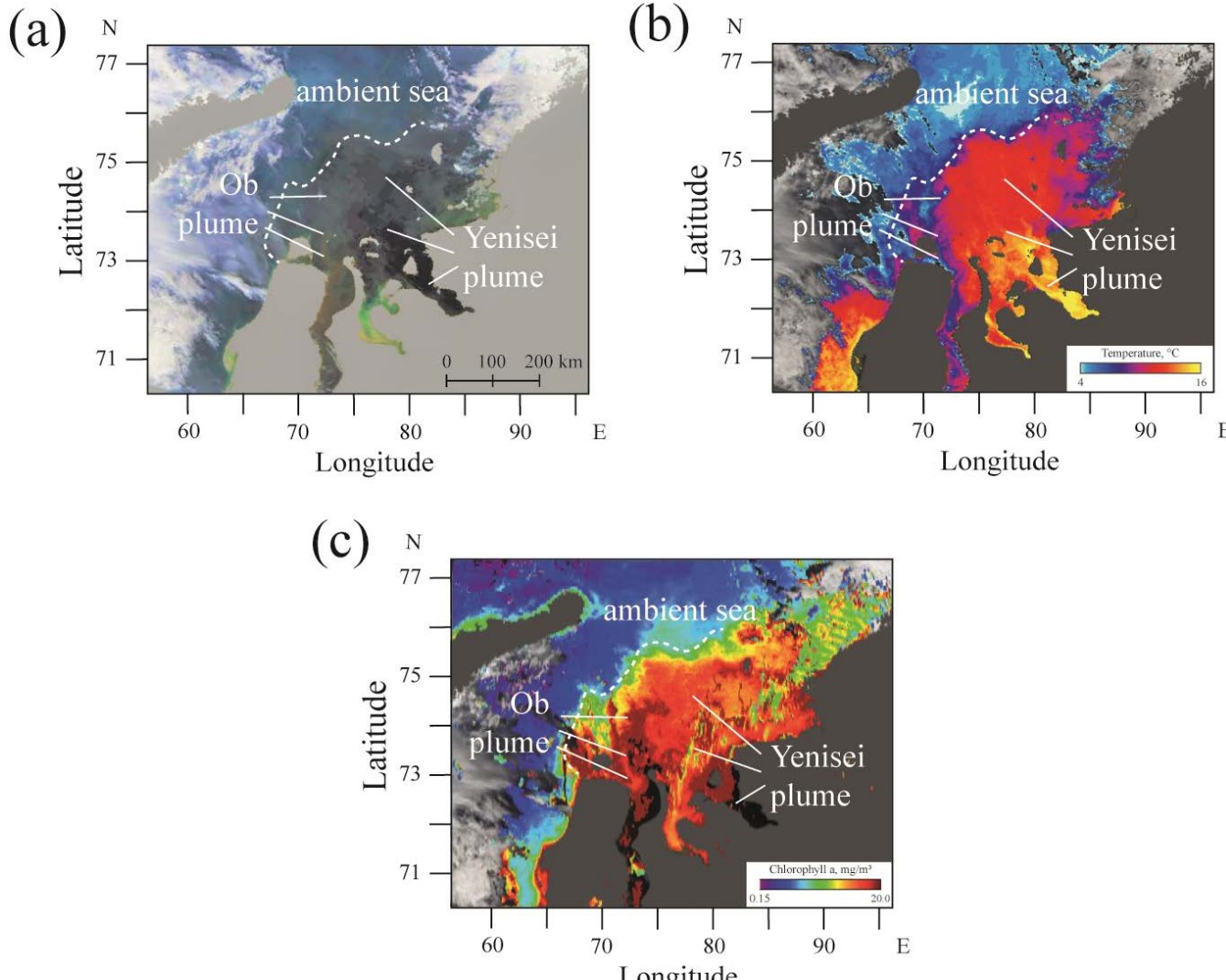

**Figure 8: Corrected reflectance (a), brightness temperature (b), and concentration of chlorophyll-a (c) from MODIS Aqua satellite image of the central part of the Kara Sea acquired on 15 July 2016.**

Unlike the Ob-Yenisei plume formed by discharges of two large rivers, the Khatanga River is the only large river that forms the buoyant plume in the southwestern part of the Laptev Sea. The Lena River is the closest large river to the Khatanga Gulf that inflows to the Laptev Sea. However, the distance between the Khatanga Gulf and the Lena Delta is > 350 km (Fig. 7b). Moreover, 80-90% of large freshwater discharge from the Lena River (530 km$^3$ annually or 16800 m$^3$/s on average) inflows to the Laptev Sea from the eastern part of the Lena Delta, while its western part accounts only for 6-10% [Fedorova et al.,

2012]. As a result, the large Lena plume is formed in the central and southeastern parts of the Laptev Sea, in particular, it

does not spread westward from the 120° E longitude to the southwestern parts of the Laptev Sea and does not merge with the Khatanga plume [Fofonova et al., 2015].

Continuous surface salinity measurements along the ship track in the southwestern part of the Laptev Sea detected the northern and northeastern segments of the border of the Khatanga plume in September 2017 (Fig. 7c). Surface salinity remained less than 25 at a distance of 200-250 km from the Khatanga Gulf and 450-500 km from the Khatanga River mouth. Another field survey conducted at the southwestern part of the Laptev Sea in the end of August 2000 also showed large spatial extents of the Khatanga plume (Fig. 7b). Distinct location of the plume boundary was not detected during this field survey due to absence of continuous measurements of surface salinity. However, the vertical profiles of salinity obtained at the hydrographic stations showed that the isohaline of 20 in the surface layer was located at a distance of 50-200 km from the Khatanga Gulf.

Satellite observations of the study area acquired in the end of August 2000 confirmed the assessment of the spreading area of the Khatanga plume based on in situ measurements at the hydrographic stations (Fig. 9). The Khatanga plume is characterized by elevated concentrations of suspended sediments and chlorophyll-a, as compared to the ambient saline sea. As a result, spreading area of the plume can be detected at cloud-free and ice-free CR and Chl-a satellite images of the study area. BT satellite products, on the other hand, are not effective for detection of the Khatanga plume due to relatively small difference between temperature in the diluted Khatanga plume and the ambient sea. Joint analysis of CR and Chl-a satellite images revealed that the border of the Khatanga plume on 23 and 25 August 2000 (a week after ice melting and 4 days before in situ measurements in the study area) was located at a distance of 50-250 km from the Khatanga Gulf which is consistent with in situ measurements. The region of elevated turbidity and elevated concentration of chlorophyll-a located eastward from the 120° E longitude and adjacent to the Lena Delta is associated with the western part of the Lena plume.

Constantly overcast sky at the southwestern part of the Laptev Sea in summer and autumn of 2017 hindered satellite observations of the Khatanga plume that could support the in situ measurements on 17-18 September 2017. However, the Khatanga plume was regularly observed at cloud-free images of the study area acquired in 2000-2019. In particular, the spreading area and the boundary of the Khatanga plume are distinctly visible at several CR and Chl-a satellite images acquired during days when the whole southwestern part of the Laptev Sea was free of clouds (Fig. 10). Satellite observation showed significant variability of position and shape of the Khatanga plume border presumably associated with variability of external wind forcing. Wind-induced resuspension of bottom sediments that regularly occurs over shallow coastal areas between the Khatanga Gulf and the Lena Delta often hinders precise detection of the southeasternmost segment of the Khatanga plume border. Nevertheless, satellite observations revealed that the Khatanga plume regularly occupied large area in the southwestern part of the Laptev Sea adjacent to the Khatanga Gulf. Maximal spatial extent of the Khatanga plume observed at satellite imagery in 2000-2019 varied between 150 and 250 km from the Khatanga Gulf.

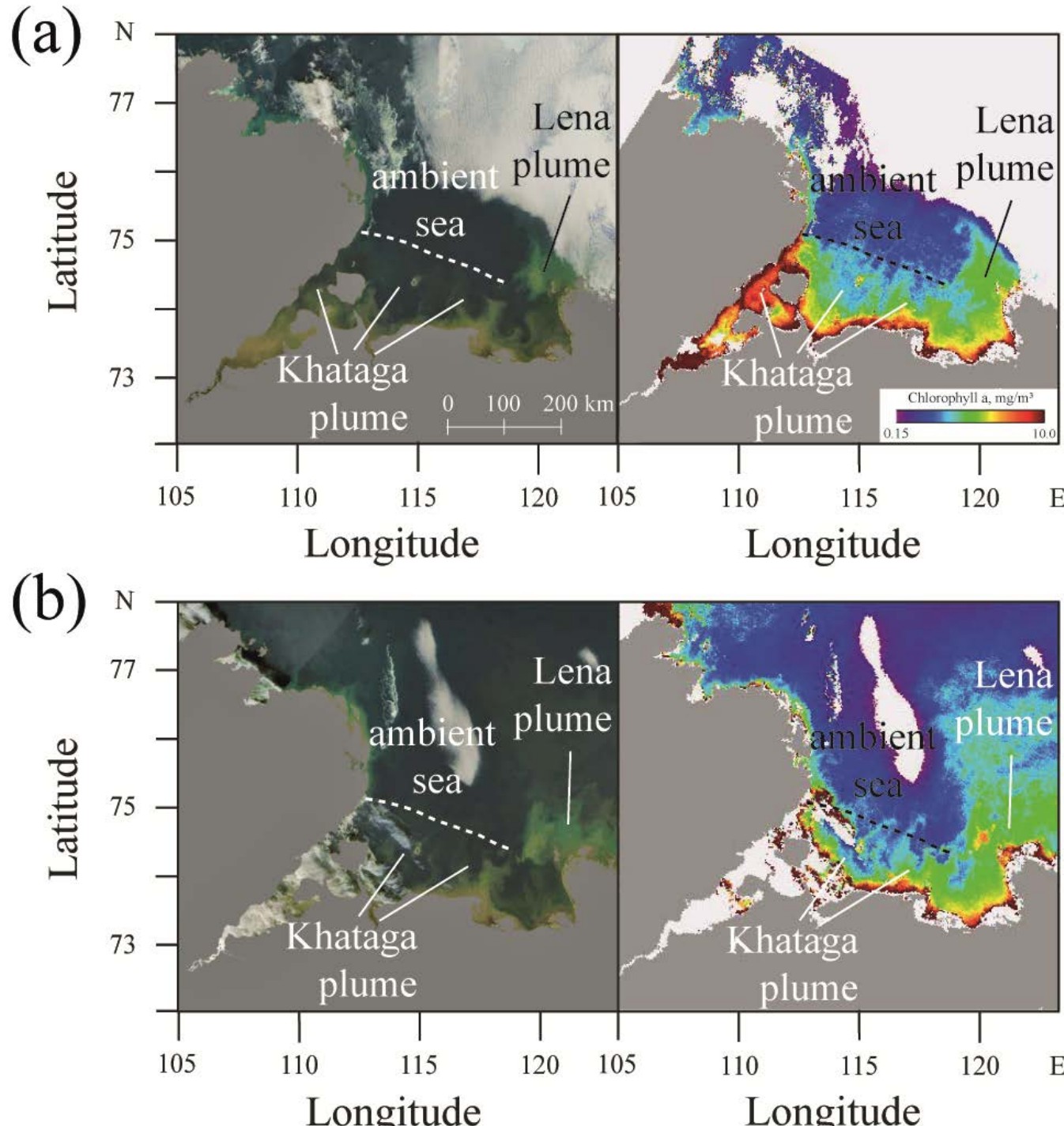

**Figure 9: Corrected reflectance (left) and concentration of chlorophyll-a (right) from MODIS Terra satellite images of the western part of the Laptev Sea acquired on 23 September (a) and 25 September (b) 2000.**

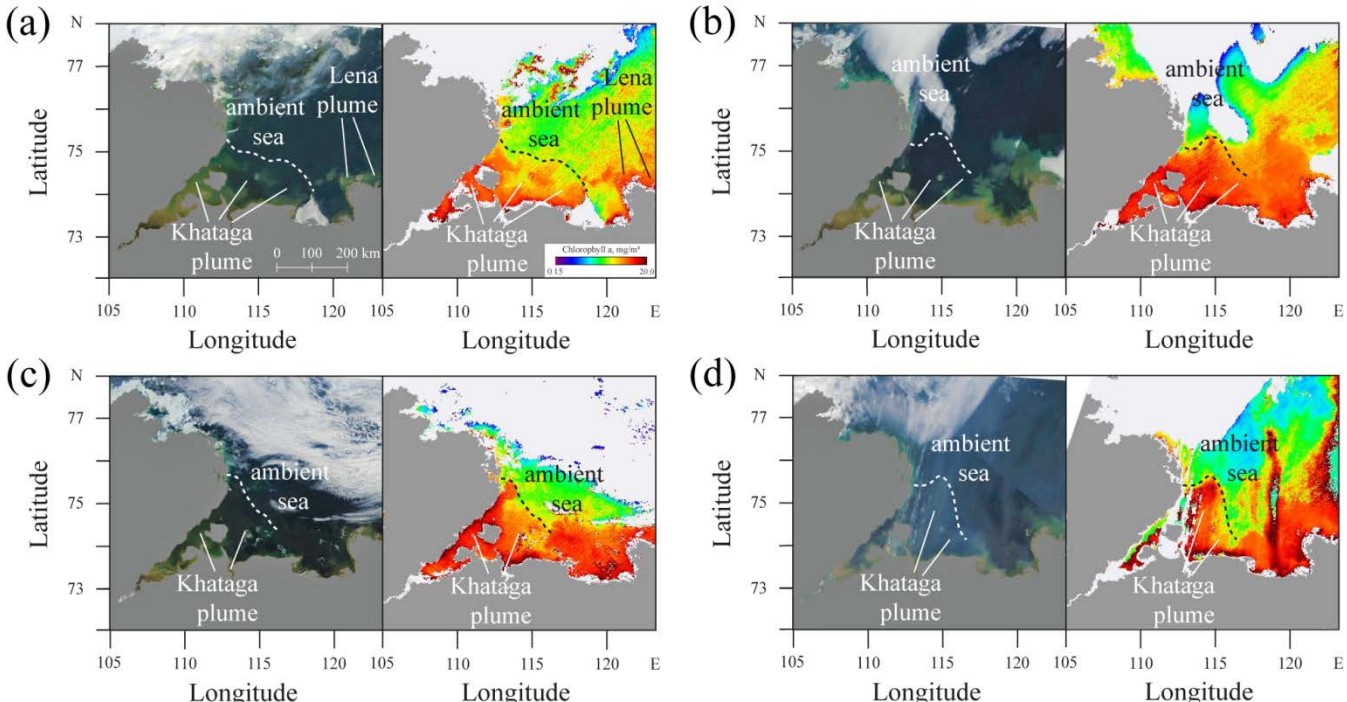

**Figure 10: Corrected reflectance (left) and concentration of chlorophyll-a (right) from MODIS Terra and MODIS Aqua satellite images of the western part of the Laptev Sea acquired on 8 August 2003 (a), 13 August 2011 (b), 8 August 2013 (c), and 3 August 2019 (d).**

## 4 Discussion

River discharge rate, wind forcing, and tidal circulation are the main factors that determine estuarine circulation and govern vertical structure and spatial scales of river plumes [Nash et al., 2009; Horner-Devine et al., 2015]. In situ measurements and satellite observation of the Yenisei and Khatanga plumes described above revealed that spatial scales of these plumes were similar, while discharge rates of the Yenisei and Khatanga rivers differed by one order of magnitude. The Yenisei and Khatanga gulfs are closely located and have similar sizes, geomorphology, and climatic conditions, albeit significantly different tidal forcing. General shelf circulation does not significantly impact buoyancy-driven and wind-driven spreading of the large Yenisei and Khatanga plumes [Guay et al., 2001; Panteleev et al., 2007; Zatsepin et al., 2010; 2015; Osadchiev et al., 2017; 2019]. Wind forcing, on the other hand, can strongly affect these river plumes in case of strong and long-duration winds. However, it is not the case of this study, because wind conditions several weeks before and during the field sampling were moderate and low (Fig. 3-4). Thus, we presume that similarity in spatial extents of the Yenisei and Khatanga plumes is caused by difference in estuarine tidal mixing forcing that determined difference in their structure, in particular, different freshwater fractions.

The estuarine dynamical and mixing regime determined by the buoyancy and tidal forcing can be characterized by two dimensionless parameters, namely, the freshwater Froude number $Fr_f = \frac{U_R}{\sqrt{\beta g S H}}$ and a mixing parameter $M = \frac{C_D U_T^2}{w N_0 H^2}$, where $U_R$ is the inflowing river velocity, $U_T$ is the amplitude of the depth-averaged estuarine tidal velocity, $w$ is the estuarine tidal frequency, $S$ is the ambient sea salinity, $H$ is the depth of an estuary, $g$ is the gravity acceleration, $\beta$ is the saline contraction coefficient prescribed equal to $7.7 \cdot 10^{-4}$, $C_D$ is the quadratic drag coefficient for wind stress parameterization prescribed equal to $10^{-3}$, $N_0 = \sqrt{\frac{\beta g S}{H}}$ is the buoyancy frequency for maximum top-to-bottom salinity variation in an estuary [Geyer and MacCready, 2014]. Values of these parameters for the Yenisei and Khatanga gulfs are the following: $Fr_f^Y \sim 0.5 / (7.7 \cdot 10^{-4} \cdot 10 \cdot 32 \cdot 10)^{0.5} \sim 0.3$, $M^Y \sim 10^{-3} \cdot (0.3)^2 / (2.3 \cdot 10^{-5} \cdot (7.7 \cdot 10^{-4} \cdot 10 \cdot 32 / 10)^{0.5} \cdot (10)^2) \sim 0.2$; $Fr_f^K \sim 0.1 / (7.7 \cdot 10^{-4} \cdot 10 \cdot 32 \cdot 10)^{0.5} \sim 0.06$, $M^K \sim 10^{-3} \cdot (0.8)^2 / (2.3 \cdot 10^{-5} \cdot (7.7 \cdot 10^{-4} \cdot 10 \cdot 32 / 10)^{0.5} \cdot (10)^2) \sim 1.8$, where the superscripts $Y$ and $K$ refer to the Yenisei and Khatanga gulfs, respectively. Therefore, according to classification of estuaries developed by Geyer and MacCready [2014], the Yenisei Gulf is a salt wedge estuary, while the Khatanga Gulf is characterized by strain-induced periodic stratification.

Large freshwater discharge of the Yenisei River estimated as 30000 m$^3$/s did not experience intense mixing in the estuary and formed a relatively shallow (8-12 m), low-salinity, strongly stratified plume (Fig. 5a). In situ and satellite measurements showed that the boundary of the Yenisei plume was located 200-250 km from the Yenisei Gulf (Fig. 7a, 8). Area of the Yenisei plume was approximately 60000 km$^2$.On the other hand, moderate discharge of the Khatanga River estimated as 3000 m$^3$/s formed deep (15-25 m), but diluted and weakly stratified plume in the middle of September 2017 (Fig. 6a). In situ measurements along the ship track showed that surface salinity in the southwestern part of the Laptev Sea remained less than 25 at a distance of 200-250 km from the Khatanga Gulf (Fig. 7c). Another field survey conducted at the southwestern part of the Laptev Sea in the end of August 2000 (Fig. 7b), as well as satellite observations of the Khatanga plume in 2000-2019 (Fig. 9, 10) confirmed that area of the Khatanga plume formed by moderate discharge of the Khatanga River varied between 30000 and 50000 km$^2$, i.e. was of the same order as the area of the Yenisei plume formed by large discharge of the Yenisei River.

This result is supported by vertical distributions of freshwater "volume" among salinity ranges in the Yenisei (Fig. 5c) and Khatanga (Fig. 6c) plumes. The majority of freshwater "volume" contained in the Yenisei plume in the inner estuary was located in 0-5 salinity range. After the plume propagated to the outer estuary, this freshwater "volume" transferred to 5-10 salinity range and remained there, when the plume was spreading over the open sea. As a result, 95% of freshwater discharge of the Yenisei River was mixed with a relatively small volume of saline sea water and formed 0-5 and 5-10 salinity ranges within the Yenisei plume with relatively small volumes. On the other hand, freshwater "volume" contained in the Khatanga plume steadily transferred from 0-5 salinity range near the river mouth to 20-25 salinity range in the outer estuary and sea shelf. Therefore, freshwater discharge of the Khatanga River was diluted by a large volume of saline sea water in the estuary and formed the 20-25 saline Khatanga plume with relatively large volume. This plume spread outside the estuary and

covered wide area in the southwestern part of the Laptev Sea. Therefore, intense estuarine tidal mixing caused formation of an anomalously deep and large Khatanga plume from relatively small discharge of the Khatanga River. In particular, depths of other large river plumes in the Arctic Ocean including the Ob-Yenisei, Lena, Indigirka, Kolyma, and Mackenzie plumes do not exceed 15 m, which is much less than the depth of the Khatanga plume (15-25 m) [Pavlov et al., 1996; Saveleva et al., 2008; 2010; Mulligan and Perry, 2019].

In order to quantify and compare volumes of saline sea water that mix with freshwater discharge in the Yenisei and Khatanga gulfs, we used the Knudsen relations $Q_{in} = S_{out} \cdot Q_{river} / (S_{in} - S_{out})$, $Q_{out} = S_{in} \cdot Q_{river} / (S_{in} - S_{out})$, where $Q_{in}$ and $Q_{out}$ are the time-averaged volume fluxes from the ocean to the estuary (inflow in the bottom layer) and from the estuary to the ocean (outflow in the surface layer), $S_{in}$ and $S_{out}$ are the related inflow and outflow salinities, $Q_{river}$ is the time-averaged discharge from the river to the estuary [Knudsen, 1900; Burchard et al., 2018]. The locations of the seaward borders of the

estuaries were selected at station 5336 at the Yenisei transect and station 5632 at the Khatanga transect. The inflow and outflow volume fluxes for the Yenisei Gulf during the period of in situ measurements in July 2016 are equal to $Q_{in}^Y \sim 8 \cdot 30000 / (32 - 8) = 10000$ m$^3$/s, $Q_{out}^Y \sim 32 \cdot 30000 / (32 - 8) = 40000$ m$^3$/s, while the related values for the Khatanga Gulf in September 2017 are equal to $Q_{in}^K \sim 22 \cdot 3000 / (32 - 22) = 6600$ m$^3$/s, $Q_{out}^K \sim 32 \cdot 3000 / (32 - 22) \sim 9600$ m$^3$/s. Therefore, according to this assessment, the discharge of the Khatanga River (3000 m$^3$/s) is mixed with twice greater volume of saline

water (6600 m$^3$/s) in the estuary, while the discharge of the Yenisei River (30000 m$^3$/s) is mixed with three times less volume of saline water (10000 m$^3$/s) in the estuary.

According to MacCready et al. [2018], we quantified the volume-integrated net mixing $N$ in the Yenisei and Khatanga gulfs using the relation $N = S_{in} \cdot S_{out} \cdot Q_{river}$. We obtain $N^Y \sim 32 \cdot 8 \cdot 30000 \sim 7.7 \cdot 10^6$ (g/kg)$^2$ m$^3$/s, $N^K \sim 32 \cdot 22 \cdot 3000 \sim 2.1 \cdot 10^6$ (g/kg)$^2$ m$^3$/s, that shows that tidally averaged net mixing in the Yenisei Gulf is several times greater than that in the Khatanga

Gulf due to significantly greater freshwater discharge to the Yenisei Gulf, as compared to the Khatanga Gulf. This difference is even more pronounced for the inner parts of these estuaries, which borders were selected at station 5341 in the Yenisei Gulf and station 5629 in the Khatanga Gulf. The obtained values of tidally averaged net mixing in the inner estuaries were equal to $N_{inner}^Y \sim 25 \cdot 5 \cdot 30000 \sim 3.7 \cdot 10^6$ (g/kg)$^2$ m$^3$/s and $N_{inner}^K \sim 15 \cdot 11 \cdot 3000 \sim 0.5 \cdot 10^6$ (g/kg)$^2$ m$^3$/s.

The long-term average mixing values $N^Y$ and $N^K$ were estimated for the Yenisei and Khatanga gulfs which seaward borders

were determined according to the morphology of the estuaries, i.e., based on geographical considerations. According to Burchard [2020], the same net mixing value occurs within the "estuaries" which outer borders were determined by isohalines of certain salinity $S_E = \sqrt{S_{in}S_{out}}$. This approach considers the estuarine mixing in terms of isohaline space rather than in Eulerian space and, therefore, reproduces the estuarine – river plume continuum. In particular, the values of these "estuarine salinity borders" are ~ 16 for the Yenisei plume and ~ 26.5 for the Khatanga plume. These "estuarine salinity borders"

extend far off the Khatanga and Yenisei gulfs to the open sea. Moreover, they generally correspond to the plume borders (determined by the salinity isohaline of 25) due to small distance between the isohalines of 16 and 25 at the central part of the Kara Sea (Fig. 5a) and between the isohalines of 25 and 26.5 at the southwestern part of the Laptev Sea (Fig. 6a). This

correspondence evidences that dynamical processes that occur in the estuaries which have relatively small areas (~10000 km$^2$) strongly influence spreading and mixing of the freshened surface layers over wide areas in the open sea.

## 430 **5 Conclusions**

In this study we address the Yenisei and Khatanga plumes formed by discharges of the large estuarine rivers to the Kara and Laptev seas. Based on tidal level and velocity data, tidal modelling, in situ measurements, and satellite observations, we demonstrate that estuarine tidal mixing conditions strongly affect vertical structure and spatial extents of the Yenisei and Khatanga plumes. Tidal gauge measurements and tidal modelling showed that tidal forcing is intense in the Khatanga Gulf,
maximal tidal velocity increases from 20–50 cm/s in the outer part of the estuary to 80–100 cm/s in its inner part. It results in intense surface-to-bottom tidal-induced mixing within the majority of the area of the Khatanga Gulf. Tidal forcing in the Yenisei Gulf, however, is relatively low (maximal tidal velocity < 25 cm/s) and caused only limited estuarine mixing in the surface layer.

In situ salinity measurements revealed significant difference in vertical salinity structure within the Yenisei and Khatanga
gulfs that induced different spreading patterns of the Yenisei and Khatanga plumes in the open sea. Freshwater discharge from the Yenisei River experienced low mixing within the Yenisei Gulf due to weak tidal circulation and large river runoff volume. It resulted in strong stratification between the freshened surface layer and the subjacent saline sea within the gulf. Therefore, large volume of freshwater discharge mixed with relatively small volume of saline sea water formed the Yenisei plume that propagated from the estuary and spread in the open sea. The low-salinity and strongly stratified Yenisei plume
had relatively small depth (< 12 m), its lateral border manifested by sharp salinity gradient was located at a distance of 200-250 km from the Yenisei Gulf and 450-500 km from the Yenisei River mouth.

Freshwater discharge from the Khatanga River, on the other hand, experienced strong tidal mixing in the Khatanga Gulf. Surface salinity dramatically increased within the gulf from 0 near the river mouth to 20 at the estuary mouth. However, surface salinity in the outer part of the Khatanga Gulf remained much smaller than salinity of water that flowed from the
open sea to the gulf in the bottom layer, i.e., river runoff volume was large enough not to be totally mixed within the estuary and to form a freshened surface layer, albeit very diluted. As a result, the Khatanga plume that propagated off the estuary to the open sea had small freshwater fraction, i.e., it was formed by relatively small volume of freshwater discharge mixed with large volume of saline water. The weakly stratified Khatanga plume was very deep (up to 25 m) and occupied large area and volume (in relation to the river runoff rate) at the Laptev Sea. In situ measurements and satellite observations showed that
lateral border of the Khatanga plume was located at a distance of 150-250 km from the Khatanga Gulf and 400-500 km from the Khatanga River mouth.

A number of previous works stated that river discharge rate and wind forcing are the main external conditions that govern size of a river plume [Whitney and Garvine, 2005; O'Donnell et al., 2008; Chant, 2011; Korotkina et al., 2011; Osadchiev and Zavialov, 2013; Korotkina et al., 2014; Horner-Devine et al., 2015]. In this study we show that influence of estuarine

mixing on spatial scales of a large river plume in a coastal sea can be of the same importance as the roles of river discharge rate and wind forcing. The Yenisei and Khatanga plumes addressed in this work evidence that plumes with similar areas can be formed by rivers with significantly different discharge rates. Analogously, we infer that rivers with similar discharge rates can form plumes with significantly different areas in case of large differences in intensity of estuarine mixing, however, the detailed analysis of this supposition is beyond the current study.

The results obtained in this study allow new insights into the processes of spreading and transformation of river discharge in coastal sea. This issue is especially important for the Arctic Ocean that receives more than three-quarters of total freshwater runoff from ten large rivers [Gordeev et al., 1996; Carmack, 2000; Guay et al., 2001]. Moreover, during freshet periods in June – July the majority of annual freshwater discharge of large Arctic rivers flows into coastal areas which are mostly covered by ice. As a result, formation and spreading of these large river plumes is only weakly affected by wind forcing

during these periods. Therefore, study of estuarine and deltaic mixing conditions that determine spatial scales and structure of large Arctic river plumes is essential for assessment of large-scale freshwater transport in the Arctic Ocean which plays a key role in stratification and ice formation, as well as in many physical, biological, and geochemical processes [Carmack et al., 2010; Nummelin et al., 2016].

**Data availability**

Tidal gauge data were downloaded from the ESIMO web repository http://portal.esimo.ru/portal (available after registration). The AOTIM5 tidal model was downloaded from the Earth & Space Research (ESR) web repository https://www.esr.org/research/polar-tide-models/list-of-polar-tide-models/aotim-5/. The ERA5 atmospheric reanalysis data were downloaded from the ECMWF web repository https://www.ecmwf.int/en/forecasts/datasets/archive-datasets/reanalysis-datasets/era5. The river discharge data were downloaded from the FSHEMR web repository http://gis.vodinfo.ru/ (available

after registration). The Terra/Aqua MODIS satellite data were downloaded from the NASA web repository https://ladsweb.modaps.eosdis.nasa.gov/. The in situ data are available in supplementary information.

**Competing interests**

The authors declare that they have no conflict of interest.

**Acknowledgments**

The authors are grateful to the editor John M. Huthnance and two anonymous reviewers for their comments and recommendations that served to improve the article. This research was funded by the Ministry of Science and Education of Russia, theme 0149-2019-0003 (collecting of in situ data); the Russian Foundation for Basic Research, research projects 18-

05-60302 (processing of in situ data), 18-05-60069 (analysis of atmospheric reanalysis data), 20-35-70039 (analysis of in situ data), 18-05-60250 (tidal analysis and modelling), and 18-05-00019 (study of river plumes); the Russian Science Foundation, research project 18-17-00089 (collecting of river discharge data); the Grant of the President of the Russian Federation for state support of young Russian scientists – candidates of science, research project MK-98.2020.5 (study of freshwater transport). The river discharge data were downloaded from the repository of the Federal Service for Hydrometeorology and Environmental Monitoring of Russia http://gis.vodinfo.ru/ (available after registration). The ERA5 reanalysis data were downloaded from the European Centre for Medium-Range Weather Forecasts (ECMWF) website https://www.ecmwf.int/en/forecasts/datasets/archive-datasets/reanalysis-datasets/era5.

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
