# Peer review of "Influence of Estuarine Tidal Mixing on Structure and Spatial Scales of Large River Plumes"

_Ocean Science, 2019_

## Referee Comment (RC1) · Anonymous Referee #1 · 12 Feb 2020

General comments

The manuscript describes using in-situ salinity observations to study the spatial scales and contrasting structures of two river plumes in the Arctic Ocean. The concept of freshwater volume is used to get new information from observational data and different vertical salinity structures in the two river plumes are demonstrated clearly. Even though one river has an order of magnitude greater discharge than the other, the limits of salinity concentrations consistent with spreading of riverine water are detected ∼500km from both river mouths. Determining the processes that control riverine flow into the ocean is important for understanding the impacts of rivers on coastal and shelf

regions. The manuscript is generally well written.

The title and abstract both mention tidal mixing as the primary process responsible for the observed differences in two river plumes. However, there is no analysis to demonstrate this in the manuscript. For the focus of the paper to be tidal mixing there needs to be some investigation of the processes involved. Another issue is that volumes and areas of the river plumes are inferred using data from one linear transect per river, but the justification for the calculations related to the width of the river plumes is not well argued (see specific comments).

Specific comments

Line 20: The assertion that "rivers with similar discharge rates can form plumes with significantly different areas" is not supported by the data presented in the manuscript.

Line 97: mean wind speed over 14 days of $\sim$7m/s is quite high and could include periods of strong wind speeds from different directions; plotting appropriate time series would give more information and show if wind forcing might impact on the river plume development.

Line 117 and figures 3b and 4b: the plotted freshwater fractions are not consistent with the definition given: eg S = 15 => F = (32 - 15) / 32 = 0.53 not 1 – 1.5% (which is plotted).

Figures 3c and 4c: How was the "total share of FV among SL" derived?

Line 128: the freshwater in different salinity layers is not a percentage volume since the changing width of the gulf is not accounted for.

Lines 168-172: This section needs clarifying. In "This result is in good agreement with" etc: what result is being referred to? The data in the manuscript is for rivers with very different discharge rates not rivers with the same discharge rate. Is figure 6 and its description based on Fischer (1972) and Nash (2009)?

[Figure]

Lines 186-194: The authors suggest that data from transects are representative of total surface areas of the river plumes because the Yenisei and Khatanga are 'large rivers' and so the plumes have similar zonal and meridional extents. However, the cited references [Pavlov et al., 1996; Zatsepin et al., 2010; Zavialov et al., 2015] show high variability in size and shape of the Yenisei and Khatanga river plumes. Also, it is possible in figure 5a) that some of the freshwater in the "Yenisei plume" comes from the nearby Ob River (Zavialov et al 2015; Osadchiev et al 2017).

Lines 220-230: the calculation of the freshwater volume. Is this just for the limits of the two gulfs, or does it include the river plumes? Do the changing widths of the gulfs and plumes impact this calculation? What about flow to the ocean through other channels (both gulfs split in two at the seaward end)? In line 228, the agreement between the ratios of freshwater volume and river discharges isn't exactly "proof" that the transects can be used to infer freshwater volume.

Web links to access the river discharge and atmospheric data used in the analysis are included but there is no information about access to the salinity observations.

Technical corrections

Line 13: exhibits -> experiences.

Line 17: delete "obtained"

Line 47: accounts to -> accounts for.

Line 58: Kowalik and Proshutinsky, 1994 is missing from the references list.

Line 100: the date "24-18 September" is wrong in the caption. The colour palette is not very effective – could omit the range 1000-1010 hPa.

Lines 110 and 139: "several meters" lacks precision.

Line 121: insert "." after 2015]

Line 125: omit "far".

Figure 5: need to label that the discharge rates are shown, and check their units. Also specify which part of the water column.

Figure 6: freshwater fraction values should be less than 1.

Line 217 was spreading -> spread

Line 235: omit "getting".

Line 313: Kulikov et al. doi reference is incomplete.

---

## Referee Comment (RC2) · Anonymous Referee #2 · 25 Feb 2020

The paper presents very interesting observations of two major river plumes in the Arctic basin. There is a paucity of such information in the oceanographic literature, so the paper certainly merits publication in the Ocean Science. However, some details of the data analysis and interpretation need improvements.

In my opinion, the authors pay too much attention to the fact that the outflows from the Yenisei Gulf and the Khatanga Gulf form plumes of roughly the same offshore extension, although the freshwater discharges of the two rivers differ by an order of magnitude ($\sim$30,000 mˆ3/s for the Yenisei River vs $\sim$3,000 mˆ3/s for the Khatanga River). According to the authors, this happens due to the different intensity of tidal

mixing in the two gulfs. I think this observation is rather trivial and obvious. Besides, it's not entirely accurate. First, the Yenisei River plume indeed separates from the coast and extends offshore (northward) over ~300 km from the estuarine mouth. The Khatanga River plume on the other hand remains attached to the Taymyr Peninsular coastline on its left flank (facing downstream) so its northward spreading cannot be characterized as the offshore extension (even more so in August 2000). Second, the wind forcing, while weak, is upwelling-favorable for the Khatanga River plume (in 2017) and is downwelling-favorable for the Yenisei River plume. The authors do not describe the wind forcing conditions prior to shipboard surveys, and the plumes of such spatial scales can keep a "memory" of the wind forcing on time scales of a week or even more if the wind is not strong. So the wind field snapshots at the time of measurements are not entirely convincing. I also somewhat disagree with the authors' interpretation of the plume structure formed by large rivers (lines 187-190, page 11): In fact, both "medium-size" and "large" (author's terminology) river plumes have the anticyclonic bulge region near the mouth and the semi-geostrophic, narrower coastal current farther downstream, as long as the Coriolis force is important. In this regard, the Amazon River and the Congo River plumes are not quite relevant since they are near the equator, while other major river plumes do have both a bulge region and a coastal current (far field), including the Mississippi plume, The Yangtze plume, the La Plata plume, the Columbia River, the Danube River, the Siberian rivers, etc.

Some minor issues with the manuscript:

Line 46 and later: I think it's better to use mˆ3 /s units for the freshwater discharge throughout the text.

Line 58: "...tidal amplitude and velocity..." Amplitude of what, perhaps the free surface? As for the velocity, is it also an amplitude or rms?

Line 86: "...performed at 100 m spatial resolution...". How can it be? I thought the water was pumped continuously under way. Do the authors imply the averaging interval

here?

Line 92: "...and 200 km far from the river mouths...". "Far" is not needed here.

Line 107:" ...Kara Sea shelf (stations 5336-5350)." The statement is misleading; it should read "stations 5333-5336 and 5349-5350".

Lines 123-124: "As a result, the majority of river runoff propagated off the estuary...". This is a somewhat strange proposition; the riverine discharge should "propagate off the estuary", otherwise there will a freshwater flux convergence in the estuary and the estuary will be continuously getting fresher.

Line 126 and below: "...was located in two salinity layers...". "Layer" is not a good choice in this context; it is one buoyant layer, just comprising different salinity classes or ranges or whatever word the authors would prefer.

Line 138: Is the salinity gradient in this context "stable" or constant?

―――――――――――――

---

## Author Comment (AC1) · 10 Apr 2020

C: General comments The manuscript describes using in-situ salinity observations to study the spatial scales and contrasting structures of two river plumes in the Arctic Ocean. The concept of freshwater volume is used to get new information from observational data and different vertical salinity structures in the two river plumes are demonstrated clearly. Even though one river has an order of magnitude greater discharge than the other, the limits of salinity concentrations consistent with spreading of riverine water are detected ∼500km from both river mouths. Determining the processes that control riverine flow into the ocean is important for understanding the im-

pacts of rivers on coastal and shelf regions. The manuscript is generally well written. The title and abstract both mention tidal mixing as the primary process responsible for the observed differences in two river plumes. However, there is no analysis to demonstrate this in the manuscript. For the focus of the paper to be tidal mixing there needs to be some investigation of the processes involved. Another issue is that volumes and areas of the river plumes are inferred using data from one linear transect per river, but the justification for the calculations related to the width of the river plumes is not well argued (see specific comments).

R: Many thanks for your important comments that served to significantly improve the article. First, we added a new Section 3.1 focused on tidal circulation and tidal mixing in the Yenisei and Khatanga gulfs. Second, we added analysis of satellite observations of the Yenisei and Khatanga plumes, as well as extended the in situ data from two linear transects to support assessments of their spatial characteristics. We described and discussed new in situ and satellite data (Section 3.4), as well as analyzed wind forcing during the extended time periods (Section 3.2) that confirmed the assessments of spatial extents of the Yenisei and Khatanga plumes.

C: Specific comments Line 20: The assertion that "rivers with similar discharge rates can form plumes with significantly different areas" is not supported by the data presented in the manuscript.

R: We agree that this statement is not supported by the data presented in the manuscript. This assertion was removed from the abstract.

C: Line 97: mean wind speed over 14 days of 7m/s is quite high and could include periods of strong wind speeds from different directions; plotting appropriate time series would give more information and show if wind forcing might impact on the river plume development.

R: According to your recommendation we added analysis of daily averaged wind speed and direction during 29 June – 26 July 2016 for the Yenisei plume and during 8 August

– 18 September 2017 for the Khatanga plume. These wind time series cover ice-free periods at the study areas from decline of ice coverage to in situ measurements in the Yenisei and Khatanga plumes, i.e., the periods when wind forcing can influence river plumes. In Section 3.2 we provide analysis of these time series and showed that speed of the considered wind forcing was mainly moderate and low. In particular, the longest observed periods of continuous moderate and strong wind (> 5 m/s) were only 4 days in the central part of the Kara Sea and 3 days in the western part of the Laptev Sea. Wind direction during the study periods was highly variable due to high variability of atmospheric pressure accompanied by multiple cyclones and anticyclones. As a result, the wind forcing averaged during 2-week time periods is characterized by even more low wind speed (< 4 m/s) for the considered periods. Therefore, we presume that the Yenisei and Khatanga plumes were only weakly affected by wind forcing during the periods preceding the in situ measurements. As a result, the registered spatial extents of the Yenisei and Khatanga plumes depend mainly on river discharge conditions and estuarine mixing. This issue was clarified in the text.

C: Line 117 and figures 3b and 4b: the plotted freshwater fractions are not consistent with the definition given: eg $S = 15 \Rightarrow F = (32 - 15) / 32 = 0.53$ not $1 – 1.5\%$ (which is plotted).

R: Many thanks for this comment. In this study we calculated the fraction $(S_0 – S) / S$, i.e., the ratio between volumes of river water $V_{river}$ and sea water $V_{sea}$ in the water parcel, not the fraction $(S_0 – S) / S_0$, i.e., the ratio between volume of river water $V_{river}$ and total volume $V_{river} + V_{sea}$ of the water parcel, as it was incorrectly written in the text. This mistake was corrected in the text, and in the related figures.

C: Figures 3c and 4c: How was the "total share of FV among SL" derived?

R: In order to assess dilution of freshwater discharge within the Yenisei plume, we defined five different salinity ranges of river plume water, namely, $0 < S < 5$, $5 < S < 10$, $10 < S < 15$, $15 < S < 20$, $20 < S < 25$, as well as the salinity range $S > 25$

for the ambient sea. Then for all vertical salinity profiles of the transect we calculated local shares of freshwater volume in water column among these salinity ranges, i.e., what percentage of total freshwater volume contained in the water column is located between the isohalines of 0 and 5 (salinity range of 0-5), between the isohalines of 5 and 10 (salinity range of 5-10), etc. (Fig. 4c). Finally, we calculated total shares of freshwater volume in water column among these salinity ranges by averaging the reconstructed local shares of freshwater volume along the transect. This clarification was added to the text.

C: Line 128: the freshwater in different salinity layers is not a percentage volume since the changing width of the gulf is not accounted for.

R: In this part of the manuscript we describe the percentage of freshwater volume among different salinity ranges along the transect and do not state that it corresponds to the percentage volumes within the whole plumes. The related clarification was added to the text.

C: Lines 168-172: This section needs clarifying. In "This result is in good agreement with" etc: what result is being referred to? The data in the manuscript is for rivers with very different discharge rates not rivers with the same discharge rate. Is figure 6 and its description based on Fischer (1972) and Nash (2009)?

R: In this study we show that river discharge rate and estuarine tidal mixing are important factors that govern depth and area of a river plume. The Khatanga plume is an example of a river plume that experience strong tidal mixing in the estuary and occupies anomalously large area and volume (in relation to the river runoff rate) in the open sea. The Yenisei plume, on the opposite, experience low tidal mixing in the estuary, it is shallow and occupies relatively small area. Therefore, we demonstrate, that rivers with significantly different discharge rates (Yenisei and Khatanga) can form plumes with similar areas due to different intensity of estuarine mixing. This fact is supported by in situ measurements and satellite observations and is the main result of our work. This

result is in a good agreement with Nash (2009) who showed that salinity and depth of a near-field plume are negatively correlated with a ratio of river discharge rate and cubed estuarine tidal velocity. We use freshwater fraction of a river plume, i.e., a ratio between volumes of river and sea water that were mixed to form a plume, as a proxy of its spatial extent, which is the main idea of the Figure 6. Therefore, we make an assumption that rivers with similar discharge rates can form river plumes with significantly different freshwater fractions and, therefore, spatial extents in case of large differences in intensity of estuarine mixing. However, the detailed analysis of this assumption is beyond the current study that was clearly stated in the text. We removed the related discussion and figure from the revised version of the manuscript.

C: Lines 186-194: The authors suggest that data from transects are representative of total surface areas of the river plumes because the Yenisei and Khatanga are 'large rivers' and so the plumes have similar zonal and meridional extents. However, the cited references [Pavlov et al., 1996; Zatsepin et al., 2010; Zavialov et al., 2015] show high variability in size and shape of the Yenisei and Khatanga river plumes. Also, it is possible in figure 5a) that some of the freshwater in the "Yenisei plume" comes from the nearby Ob River (Zavialov et al 2015; Osadchiev et al 2017).

R: We agree that the data from individual transects is not enough convincing for analysis of areas of the Yenisei and Khatanga plumes. Therefore, we processed and analyzed satellite observations of the Yenisei and Khatanga plumes to reconstruct their spreading areas. Also, based on satellite data, we distinguished the Ob and Yenisei plumes within the joint Ob-Yenisei plume in the central part of the Kara Sea. Based on joint analysis of satellite and in situ data, we detected spreading areas of the Yenisei and Khatanga plumes and show that, first, spatial extents of the Yenisei and Khatanga plumes were similar during the periods of field measurements, and, second, large spreading area of the Khatanga plume was regularly registered at cloud-free satellite imagery acquired in 2000-2019.

C: Lines 220-230: the calculation of the freshwater volume. Is this just for the limits

of the two gulfs, or does it include the river plumes? Do the changing widths of the gulfs and plumes impact this calculation? What about flow to the ocean through other channels (both gulfs split in two at the seaward end)? In line 228, the agreement between the ratios of freshwater volume and river discharges isn't exactly "proof" that the transects can be used to infer freshwater volume.

R: We agree that calculation of these freshwater volume are not enough convincing due to changing widths of the gulfs, variability of salinity across the gulfs, and presence of shallow, but wide secondary channels that connect these gulfs with the sea. We omitted this paragraph from the revised version of the manuscript.

C: Web links to access the river discharge and atmospheric data used in the analysis are included but there is no information about access to the salinity observations.

R: According to your recommendation, we supplementary information with in situ data used in the study.

C: Technical corrections Line 13: exhibits -> experiences. Line 17: delete "obtained" Line 47: accounts to -> accounts for. Line 58: Kowalik and Proshutinsky, 1994 is missing from the references list. Line 100: the date "24-18 September" is wrong in the caption. The colour palette is not very effective – could omit the range 1000-1010 hPa. Lines 110 and 139: "several meters" lacks precision. Line 121: insert "." after 2015] Line 125: omit "far". Figure 5: need to label that the discharge rates are shown, and check their units. Also specify which part of the water column. Figure 6: freshwater fraction values should be less than 1. Line 217 was spreading -> spread Line 235: omit "getting". Line 313: Kulikov et al. doi reference is incomplete.

R: Thank you for these minor comments, we made the related corrections in the text.

---

## Author Comment (AC2) · 10 Apr 2020

C: The paper presents very interesting observations of two major river plumes in the Arctic basin. There is a paucity of such information in the oceanographic literature, so the paper certainly merits publication in the Ocean Science. However, some details of the data analysis and interpretation need improvements. In my opinion, the authors pay too much attention to the fact that the outflows from the Yenisei Gulf and the Khatanga Gulf form plumes of roughly the same offshore extension, although the freshwater discharges of the two rivers differ by an order of magnitude ($\sim$30,000 m3/s for the Yenisei River vs $\sim$3,000 m3/s for the Khatanga River). According to the authors, this happens

due to the different intensity of tidal mixing in the two gulfs. I think this observation is rather trivial and obvious. Besides, it's not entirely accurate. First, the Yenisei River plume indeed separates from the coast and extends offshore (northward) over 300 km from the estuarine mouth. The Khatanga River plume on the other hand remains attached to the Taymyr Peninsular coastline on its left flank (facing downstream) so its northward spreading cannot be characterized as the offshore extension (even more so in August 2000).

R: Thank you for this important comment. We agree that the data from individual transects is not enough convincing for analysis of areas of the Yenisei and Khatanga plumes. Therefore, we provided analysis of satellite observations of the Yenisei and Khatanga plumes in the revised version of the manuscript. Based on joint analysis of satellite and in situ data, we detected spreading areas of the Yenisei and Khatanga plumes and validated them against in situ measurements. We showed that, first, spatial extents of the Yenisei and Khatanga plumes were similar during the periods of field measurements, and, second, large spreading area of the Khatanga plume was regularly registered at cloud-free satellite imagery acquired in 2000-2019.

C: Second, the wind forcing, while weak, is upwelling-favorable for the Khatanga River plume (in 2017) and is downwelling-favorable for the Yenisei River plume. The authors do not describe the wind forcing conditions prior to shipboard surveys, and the plumes of such spatial scales can keep a "memory" of the wind forcing on time scales of a week or even more if the wind is not strong. So the wind field snapshots at the time of measurements are not entirely convincing.

R: According to your recommendation we added analysis of daily averaged wind speed and direction during 29 June – 26 July 2016 for the Yenisei plume and during 8 August – 18 September 2017 for the Khatanga plume. These wind time series cover ice-free periods at the study areas from decline of ice coverage to in situ measurements in the Yenisei and Khatanga plumes, i.e., the periods when wind forcing can influence river plumes. In Section 3.2 we provide analysis of these time series and showed that

speed of the considered wind forcing was mainly moderate and low. In particular, the longest observed periods of continuous moderate and strong wind (> 5 m/s) were only 4 days in the central part of the Kara Sea and 3 days in the western part of the Laptev Sea. Wind direction during the study periods was highly variable due to high variability of atmospheric pressure accompanied by multiple cyclones and anticyclones. As a result, the wind forcing averaged during 2-week time periods is characterized by even more low wind speed (< 4 m/s) for the considered periods. Therefore, we presume that the Yenisei and Khatanga plumes were only weakly affected by wind forcing during the periods preceding the in situ measurements. As a result, the registered spatial extents of the Yenisei and Khatanga plumes depend mainly on river discharge conditions and estuarine mixing. This issue was clarified in the text.

C: I also somewhat disagree with the authors' interpretation of the plume structure formed by large rivers (lines 187-190, page 11): In fact, both "medium-size" and "large" (author's terminology) river plumes have the anticyclonic bulge region near the mouth and the semi-geostrophic, narrower coastal current farther downstream, as long as the Coriolis force is important. In this regard, the Amazon River and the Congo River plumes are not quite relevant since they are near the equator, while other major river plumes do have both a bulge region and a coastal current (far field), including the Mississippi plume, The Yangtze plume, the La Plata plume, the Columbia River, the Danube River, the Siberian rivers, etc.

R: We totally agree that large river plumes have asymmetric shapes that result in their different cross-shore and alongshore extents. However, in this part of the manuscript we expressed the idea that cross-shore extents of large river plumes near their estuaries are more stable that those of small river plumes. Anyway, in the revised version of the manuscript we omitted this statement and the related discussion because we quantified the similarity of areas of the Yenisei and Khatanga plumes based on satellite observations described in Section 3.4.

C: Some minor issues with the manuscript: Line 46 and later: I think it's better to

use m3 /s units for the freshwater discharge throughout the text. Line 58: ": : :tidal amplitude and velocity: : :" Amplitude of what, perhaps the free surface? As for the velocity, is it also an amplitude or rms? Line 86: ": : :performed at 100 m spatial resolution: : :". How can it be? I thought the water was pumped continuously under way. Do the authors imply the averaging interval here? Line 92: ": : :and 200 km far from the river mouths: : :". "Far" is not needed here. Line 107:" : : :Kara Sea shelf (stations 5336-5350)." The statement is misleading; it should read "stations 5333-5336 and 5349-5350". Lines 123-124: "As a result, the majority of river runoff propagated off the estuary: : :". This is a somewhat strange proposition; the riverine discharge should "propagate off the estuary", otherwise there will a freshwater flux convergence in the estuary and the estuary will be continuously getting fresher. Line 126 and below: ": : :was located in two salinity layers: : :". "Layer" is not a good choice in this context; it is one buoyant layer, just comprising different salinity classes or ranges or whatever word the authors would prefer. Line 138: Is the salinity gradient in this context "stable" or constant?

R: Thank you for these minor comments, we made the related corrections in the text.

---

## Author Comment (AC3) · 10 Apr 2020

**Response to reviewers' comments on Osadchiev et al., 'Influence of Estuarine Tidal Mixing on Structure and Spatial Scales of Large River Plumes' – original comments are in black, responses in blue.**

*Anonymous Referee #1*

*General comments*

*The manuscript describes using in-situ salinity observations to study the spatial scales and contrasting structures of two river plumes in the Arctic Ocean. The concept of freshwater volume is used to get new information from observational data and different vertical salinity structures in the two river plumes are demonstrated clearly. Even though one river has an order of magnitude greater discharge than the other, the limits of salinity concentrations consistent with spreading of riverine water are detected ~500km from both river mouths. Determining the processes that control riverine flow into the ocean is important for understanding the impacts of rivers on coastal and shelf regions. The manuscript is generally well written.*

*The title and abstract both mention tidal mixing as the primary process responsible for the observed differences in two river plumes. However, there is no analysis to demonstrate this in the manuscript. For the focus of the paper to be tidal mixing there needs to be some investigation of the processes involved. Another issue is that volumes and areas of the river plumes are inferred using data from one linear transect per river, but the justification for the calculations related to the width of the river plumes is not well argued (see specific comments).*

Many thanks for your important comments that served to significantly improve the article. First, we added a new Section 3.1 focused on tidal circulation and tidal mixing in the Yenisei and Khatanga gulfs. Second, we added analysis of satellite observations of the Yenisei and Khatanga plumes, as well as extended the in situ data from two linear transects to support assessments of their spatial characteristics. We described and discussed new in situ and satellite data (Section 3.4), as well as analyzed wind forcing during the extended time periods (Section 3.2) that confirmed the assessments of spatial extents of the Yenisei and Khatanga plumes.

*Specific comments*

*Line 20: The assertion that "rivers with similar discharge rates can form plumes with significantly different areas" is not supported by the data presented in the manuscript.*

We agree that this statement is not supported by the data presented in the manuscript. This assertion was removed from the abstract.

*Line 97: mean wind speed over 14 days of 7m/s is quite high and could include periods of strong wind speeds from different directions; plotting appropriate time series would give more information and show if wind forcing might impact on the river plume development.*

According to your recommendation we added analysis of daily averaged wind speed and direction during 29 June – 26 July 2016 for the Yenisei plume and during 8 August – 18

September 2017 for the Khatanga plume. These wind time series cover ice-free periods at the study areas from decline of ice coverage to in situ measurements in the Yenisei and Khatanga plumes, i.e., the periods when wind forcing can influence river plumes. In Section 3.2 we provide analysis of these time series and showed that speed of the considered wind forcing was mainly moderate and low. In particular, the longest observed periods of continuous moderate and strong wind (> 5 m/s) were only 4 days in the central part of the Kara Sea and 3 days in the western part of the Laptev Sea. Wind direction during the study periods was highly variable due to high variability of atmospheric pressure accompanied by multiple cyclones and anticyclones. As a result, the wind forcing averaged during 2-week time periods is characterized by even more low wind speed (< 4 m/s) for the considered periods. Therefore, we presume that the Yenisei and Khatanga plumes were only weakly affected by wind forcing during the periods preceding the in situ measurements. As a result, the registered spatial extents of the Yenisei and Khatanga plumes depend mainly on river discharge conditions and estuarine mixing. This issue was clarified in the text.

*Line 117 and figures 3b and 4b: the plotted freshwater fractions are not consistent with the definition given: eg S = 15 => F = (32 - 15) / 32 = 0.53 not 1 – 1.5% (which is plotted).*

Many thanks for this comment. In this study we calculated the fraction $(S_0 – S) / S$, i.e., the ratio between volumes of river water $V_{river}$ and sea water $V_{sea}$ in the water parcel, not the fraction $(S_0 – S) / S_0$, i.e., the ratio between volume of river water $V_{river}$ and total volume $V_{river} + V_{sea}$ of the water parcel, as it was incorrectly written in the text. This mistake was corrected in the text, and in the related figures.

*Figures 3c and 4c: How was the "total share of FV among SL" derived?*

In order to assess dilution of freshwater discharge within the Yenisei plume, we defined five different salinity ranges of river plume water, namely, $0 < S < 5$, $5 < S < 10$, $10 < S < 15$, $15 < S < 20$, $20 < S < 25$, as well as the salinity range $S > 25$ for the ambient sea. Then for all vertical salinity profiles of the transect we calculated local shares of freshwater volume in water column among these salinity ranges, i.e., what percentage of total freshwater volume contained in the water column is located between the isohalines of 0 and 5 (salinity range of 0-5), between the isohalines of 5 and 10 (salinity range of 5-10), etc. (Fig. 4c). Finally, we calculated total shares of freshwater volume in water column among these salinity ranges by averaging the reconstructed local shares of freshwater volume along the transect. This clarification was added to the text.

*Line 128: the freshwater in different salinity layers is not a percentage volume since the changing width of the gulf is not accounted for.*

In this part of the manuscript we describe the percentage of freshwater volume among different salinity ranges along the transect and do not state that it corresponds to the percentage volumes within the whole plumes. The related clarification was added to the text.

*Lines 168-172: This section needs clarifying. In "This result is in good agreement with" etc: what result is being referred to? The data in the manuscript is for rivers with very different discharge rates not rivers with the same discharge rate. Is figure 6 and its description based on Fischer (1972) and Nash (2009)?*

In this study we show that river discharge rate and estuarine tidal mixing are important factors that govern depth and area of a river plume. The Khatanga plume is an example of a river plume that experience strong tidal mixing in the estuary and occupies anomalously large area and volume (in relation to the river runoff rate) in the open sea. The Yenisei plume, on the opposite, experience low tidal mixing in the estuary, it is shallow and occupies relatively small area. Therefore, we demonstrate, that rivers with significantly different discharge rates (Yenisei and Khatanga) can form plumes with similar areas due to different intensity of estuarine mixing. This fact is supported by in situ measurements and satellite observations and is the main result of our work. This result is in a good agreement with Nash (2009) who showed that salinity and depth of a near-field plume are negatively correlated with a ratio of river discharge rate and cubed estuarine tidal velocity. We use freshwater fraction of a river plume, i.e., a ratio between volumes of river and sea water that were mixed to form a plume, as a proxy of its spatial extent, which is the main idea of the Figure 6. Therefore, we make an assumption that rivers with similar discharge rates can form river plumes with significantly different freshwater fractions and, therefore, spatial extents in case of large differences in intensity of estuarine mixing. However, the detailed analysis of this assumption is beyond the current study that was clearly stated in the text. We removed the related discussion and figure from the revised version of the manuscript.

*Lines 186-194: The authors suggest that data from transects are representative of total surface areas of the river plumes because the Yenisei and Khatanga are 'large rivers' and so the plumes have similar zonal and meridional extents. However, the cited references [Pavlov et al., 1996; Zatsepin et al., 2010; Zavialov et al., 2015] show high variability in size and shape of the Yenisei and Khatanga river plumes. Also, it is possible in figure 5a) that some of the freshwater in the "Yenisei plume" comes from the nearby Ob River (Zavialov et al 2015; Osadchiev et al 2017).*

We agree that the data from individual transects is not enough convincing for analysis of areas of the Yenisei and Khatanga plumes. Therefore, we processed and analyzed satellite observations of the Yenisei and Khatanga plumes to reconstruct their spreading areas. Also, based on satellite data, we distinguished the Ob and Yenisei plumes within the joint Ob-Yenisei plume in the central part of the Kara Sea. Based on joint analysis of satellite and in situ data, we detected spreading areas of the Yenisei and Khatanga plumes and show that, first, spatial extents of the Yenisei and Khatanga plumes were similar during the periods of field measurements, and, second, large spreading area of the Khatanga plume was regularly registered at cloud-free satellite imagery acquired in 2000-2019.

*Lines 220-230: the calculation of the freshwater volume. Is this just for the limits of the two gulfs, or does it include the river plumes? Do the changing widths of the gulfs and plumes impact this calculation? What about flow to the ocean through other channels (both gulfs split in two at the seaward end)? In line 228, the agreement between the ratios of freshwater volume and river discharges isn't exactly "proof" that the transects can be used to infer freshwater volume.*

We agree that calculation of these freshwater volume are not enough convincing due to changing widths of the gulfs, variability of salinity across the gulfs, and presence of shallow, but wide secondary channels that connect these gulfs with the sea. We omitted this paragraph from the revised version of the manuscript.

*Web links to access the river discharge and atmospheric data used in the analysis are included but there is no information about access to the salinity observations.*

According to your recommendation, we supplementary information with in situ data used in the study.

*Technical corrections*

*Line 13: exhibits -> experiences.*

*Line 17: delete "obtained"*

*Line 47: accounts to -> accounts for.*

*Line 58: Kowalik and Proshutinsky, 1994 is missing from the references list.*

*Line 100: the date "24-18 September" is wrong in the caption. The colour palette is not very effective – could omit the range 1000-1010 hPa.*

*Lines 110 and 139: "several meters" lacks precision.*

*Line 121: insert "." after 2015]*

*Line 125: omit "far".*

*Figure 5: need to label that the discharge rates are shown, and check their units. Also specify which part of the water column.*

*Figure 6: freshwater fraction values should be less than 1.*

*Line 217 was spreading -> spread*

*Line 235: omit "getting".*

*Line 313: Kulikov et al. doi reference is incomplete.*

Thank you for these minor comments, we made the related corrections in the text.

*Anonymous Referee #2*

*The paper presents very interesting observations of two major river plumes in the Arctic basin. There is a paucity of such information in the oceanographic literature, so the paper certainly merits publication in the Ocean Science. However, some details of the data analysis and interpretation need improvements.*

*In my opinion, the authors pay too much attention to the fact that the outflows from the Yenisei Gulf and the Khatanga Gulf form plumes of roughly the same offshore extension, although the freshwater discharges of the two rivers differ by an order of magnitude (~30,000 m^3/s for the Yenisei River vs ~3,000 m^3/s for the Khatanga River). According to the authors, this happens due to the different intensity of tidal mixing in the two gulfs. I think this observation is rather trivial and obvious. Besides, it's not entirely accurate.*

*First, the Yenisei River plume indeed separates from the coast and extends offshore (northward) over 300 km from the estuarine mouth. The Khatanga River plume on the other hand remains attached to the Taymyr Peninsular coastline on its left flank (facing downstream) so its northward spreading cannot be characterized as the offshore extension (even more so in August 2000).*

Thank you for this important comment. We agree that the data from individual transects is not enough convincing for analysis of areas of the Yenisei and Khatanga plumes. Therefore, we provided analysis of satellite observations of the Yenisei and Khatanga plumes in the revised version of the manuscript. Based on joint analysis of satellite and in situ data, we detected spreading areas of the Yenisei and Khatanga plumes and validated them against in situ measurements. We showed that, first, spatial extents of the Yenisei and Khatanga plumes were similar during the periods of field measurements, and, second, large spreading area of the Khatanga plume was regularly registered at cloud-free satellite imagery acquired in 2000-2019.

*Second, the wind forcing, while weak, is upwelling-favorable for the Khatanga River plume (in 2017) and is downwelling-favorable for the Yenisei River plume. The authors do not describe the wind forcing conditions prior to shipboard surveys, and the plumes of such spatial scales can keep a "memory" of the wind forcing on time scales of a week or even more if the wind is not strong. So the wind field snapshots at the time of measurements are not entirely convincing.*

According to your recommendation we added analysis of daily averaged wind speed and direction during 29 June – 26 July 2016 for the Yenisei plume and during 8 August – 18 September 2017 for the Khatanga plume. These wind time series cover ice-free periods at the study areas from decline of ice coverage to in situ measurements in the Yenisei and Khatanga plumes, i.e., the periods when wind forcing can influence river plumes. In Section 3.2 we provide analysis of these time series and showed that speed of the considered wind forcing was mainly moderate and low. In particular, the longest observed periods of continuous moderate and strong wind (> 5 m/s) were only 4 days in the central part of the Kara Sea and 3 days in the western part of the Laptev Sea. Wind direction during the study periods was highly variable due to high variability of atmospheric pressure accompanied by multiple cyclones and anticyclones. As a result, the wind forcing averaged during 2-week time periods is characterized by even more low wind speed (< 4 m/s) for the considered periods. Therefore, we presume that the Yenisei and Khatanga plumes were only weakly affected by wind forcing during the periods preceding the in situ measurements. As a result, the registered spatial extents of the Yenisei and Khatanga plumes

depend mainly on river discharge conditions and estuarine mixing. This issue was clarified in the text.

*I also somewhat disagree with the authors' interpretation of the plume structure formed by large rivers (lines 187-190, page 11): In fact, both "medium-size" and "large" (author's terminology) river plumes have the anticyclonic bulge region near the mouth and the semi-geostrophic, narrower coastal current farther downstream, as long as the Coriolis force is important. In this regard, the Amazon River and the Congo River plumes are not quite relevant since they are near the equator, while other major river plumes do have both a bulge region and a coastal current (far field), including the Mississippi plume, The Yangtze plume, the La Plata plume, the Columbia River, the Danube River, the Siberian rivers, etc.*

We totally agree that large river plumes have asymmetric shapes that result in their different cross-shore and alongshore extents. However, in this part of the manuscript we expressed the idea that cross-shore extents of large river plumes near their estuaries are more stable that those of small river plumes. Anyway, in the revised version of the manuscript we omitted this statement and the related discussion because we quantified the similarity of areas of the Yenisei and Khatanga plumes based on satellite observations described in Section 3.4.

*Some minor issues with the manuscript:*

*Line 46 and later: I think it's better to use m^3 /s units for the freshwater discharge throughout the text.*

*Line 58: ": : :tidal amplitude and velocity: : :" Amplitude of what, perhaps the free surface? As for the velocity, is it also an amplitude or rms?*

*Line 86: ": : :performed at 100 m spatial resolution: : :". How can it be? I thought the water was pumped continuously under way. Do the authors imply the averaging interval here?*

*Line 92: ": : :and 200 km far from the river mouths: : :". "Far" is not needed here.*

*Line 107:" : : :Kara Sea shelf (stations 5336-5350)." The statement is misleading; it should read "stations 5333-5336 and 5349-5350".*

*Lines 123-124: "As a result, the majority of river runoff propagated off the estuary: : :". This is a somewhat strange proposition; the riverine discharge should "propagate off the estuary", otherwise there will a freshwater flux convergence in the estuary and the estuary will be continuously getting fresher.*

*Line 126 and below: ": : :was located in two salinity layers: : :". "Layer" is not a good choice in this context; it is one buoyant layer, just comprising different salinity classes or ranges or whatever word the authors would prefer.*

*Line 138: Is the salinity gradient in this context "stable" or constant?*

Thank you for these minor comments, we made the related corrections in the text.

---

## Author Response (AR2)

We appreciate comments and suggestions from the editor and two anonymous reviewers that served to improve the article. We adopted all these comments, below is the marked version of the manuscript.

[revised manuscript text omitted]